



# Seasonal patterns and diagnostic values of δ²H, δ¹⁸O, d-excess, and Δ¹⁷O in precipitation over Seoul, South Korea (2016–2020)

Songyi Kim[1,2], Yeongcheol Han[2], Hyejung Jung[1], Jeonghoon Lee[1]

[1]Department of Science Education, Ewha Womans University, Seoul, 03760, Republic of Korea
[2]Division of Glacier & Earth Sciences, Incheon, 21190, Korea Polar Research Institute, Republic of Korea

*Correspondence to*: Jeonghoon Lee (jeonghoon.d.lee@gmail.com)

**Abstract.** Precipitation stable isotopes are critical tracers for understanding climate variabilities and the hydrological cycles, as they enable the tracing of moisture sources, air mass mixing, and evaporation-condensation mechanisms. In mid-latitude regions such as South Korea, which are influenced by tropical and extratropical circulation, highly resolved and long-term isotope records remain scarce. Here, we analyze stable isotopes in precipitation collected bi-weekly in Seoul, South Korea, from 2016 to 2020. The oxygen isotope ratios (δ¹⁸O) ranged widely from 1.15 to –18.21‰, deuterium (δ²H) ratios varied from 3.3 to –132.0‰, and the ¹⁷O-excess ranged from 69 to –28‰. All three primary isotopes exhibited a coherent sinusoidal seasonal cycle, with the most depleted values in winter, gradual enrichment through spring, and sharp depletion during the summer monsoon, reflecting the combined influence of temperature and the amount effect. The deuterium excess (d-excess) was highest during cold, dry months and lowest in humid, rainy months, reflecting shifts in relative humidity and kinetic fractionation. Meanwhile, ¹⁷O-excess (Δ¹⁷O) exhibited a similar season trend with a smaller amplitude, suggesting that, beyond its known dependence on relative humidity and kinetic fractionation, it is also modulated by large-scale transport and water vapor mixing. The local meteoric water line closely matches the global line but winter samples show a higher intercept and a slightly steeper δ¹⁷O–δ¹⁸O slope, suggesting enhanced kinetic fractionation under continental air masses. A consistently negative δ¹⁸O–Δ¹⁷O relationship was observed except in winter when it weakened. This integrated analysis of δ¹⁸O, d-excess, and Δ¹⁷O provides a comprehensive picture of source humidity, transport dynamics, and seasonal precipitation processes in a mid-latitude East Asia, and offers a valuable reference for refining isotope-enabled climate models over East Asia.



## 1 Introduction

Global climate change has modified the hydrological cycle and increased the frequency of extreme weather events such as droughts and floods (Masson-Delmotte et al., 2021; Trenberth, 2011; Lee et al., 2011). In particular, Asia has experienced substantial changes in precipitation intensity and distribution over recent decades, coinciding with continuous surface temperature rise (Masson-Delmotte et al., 2021; Lee et al., 2016). Therefore, understanding precipitation processes, which form a critical link between the climate system and water resource management (Masson-Delmotte et al., 2021; Trenberth, 2011), is essential. In this context, stable isotopes ($\delta^{18}$O and $\delta^2$H) in precipitation have emerged as powerful tracers of atmospheric water cycling and climate dynamics (Araguás-Araguás et al., 1998; Craig, 1961; Craig and Gordon, 1965; Dansgaard, 1964; Gat, 1996). In recent years, the importance of high-resolution, long-term isotope datasets have been increasingly recognized for understanding precipitation characteristics and isotopic responses in East Asia (Chen et al., 2023; Lin et al., 2024) as they are sensitive to isotopic fractionation that occurs during phase changes such as evaporation and condensation (Bowen et al., 2018; Cappa et al., 2003; Majoube, 1971).

The stable isotopic compositions of precipitation are strongly influenced by isotopic fractionation during phase changes of water such as evaporation, condensation, and precipitation formation. Heavier isotopes ($^{18}$O and $^2$H) are preferentially removed or enriched, depending on the atmospheric conditions, and the strength of this fractionation varies with environmental parameters such as temperature, relative humidity, and precipitation amount (Conroy et al., 2016; Craig and Gordon, 1965; Gat, 1996). Two well-known relationships, the temperature effect, where colder temperatures lead to lower $\delta^{18}$O and $\delta^2$H values, and the amount effect, where increased rainfall results in isotope depletion of stable water isotope, have been widely observed in various climate regimes (Araguás-Araguás et al., 1998; Dansgaard, 1964; Lee et al., 2015). These relationships have established $\delta^{18}$O and $\delta^2$H as powerful diagnostic tools for hydrological and climatological studies, as well as paleoclimate reconstructions (Jouzel et al., 1997; Winkler et al., 2012). However, since $\delta^{18}$O and $\delta^2$H primarily reflect equilibrium fractionation processes, they often have limited utility for capturing non-equilibrium phenomena such as sub-cloud evaporation, water vapor mixing, or supersaturation during cloud formation (Gat, 1996). To address these limitations, second-order isotopic parameters have been introduced, such as deuterium excess (d-excess; $d-excess(‰) = \delta^2H - 8 \times \delta^{18}O$; defined by Dansgaard, 1964) and $^{17}$O-excess ($\Delta^{17}$O; $\Delta^{17}O = \ln(\delta^{17}O + 1) - 0.528 \times \ln(\delta^{18}O + 1)$; defined by Luz and Barkan, 2010). The d-excess reflects kinetic fractionation during oceanic evaporation and is sensitive to the conditions at the moisture source, such as relative humidity and sea surface temperature (Dansgaard, 1964; Uemura et al., 2008). Meanwhile, $\Delta^{17}$O, defined as the logarithmic deviation from the global meteoric water line between $\delta^{17}$O and $\delta^{18}$O, responds to non-equilibrium processes such as water vapor mixing and supersaturated condensation and provides unique information about the dynamical history of atmospheric moisture (Barkan and Luz, 2007; Benetti et al., 2014; Landais et al., 2008; Nyamgerel et al., 2021). Recent analytical advances have enabled high-precision $\Delta^{17}$O measurements, making this parameter a promising tracer of kinetic isotopic effects in the atmosphere. However, observational datasets for $\Delta^{17}$O remain scarce in mid-latitude East Asia, and few



studies have explored its co-variability with d-excess in this region. This study addresses these gaps by analyzing d-excess and $\Delta^{17}O$ in mid-latitude precipitation to better constrain the seasonal behavior and origin of precipitation over the Korean Peninsula.

Mid-latitude regions such as the Korean Peninsula exemplify the complex climatic controls on precipitation isotopes (Ha et al., 2012; Huang et al., 2007; Kim et al., 2019; J. Lee et al., 2013, K. Lee et al., 2003). The peninsula lies at the convergence between extratropical westerlies and the East Asian monsoon, resulting in large seasonal differences in moisture sources. In summer, moisture-rich air masses from subtropical oceans (e.g., the western North Pacific) dominate, bringing heavy monsoonal rainfall. This often yields an amount-effect signal where $\delta^{18}O$ values decrease during periods of high
precipitation (Lee et al., 2011, 2015). In contrast, winter precipitation is dominated by cold, dry continental air from the Siberian-Mongolian High, which acquires limited moisture while crossing the Yellow Sea. As a result, winter precipitation is typically more $\delta^{18}O$-depleted due to the lower temperatures and higher upstream rainout. These seasonal contrasts produce a strong annual cycle in precipitation isotope composition in the region, which is highly diagnostic of monsoon strength, moisture source changes, and temperature variability. Accordingly, long-term changes in precipitation isotopes (e.g., $\delta^{18}O$,
$\delta^2H$, and $\Delta^{17}O$) may indicate broader hydrological and climatic changes. $\Delta^{17}O$ measurements offer additional diagnostic value as they are largely temperature-independent, reflecting kinetic fractionation sensitive to the relative humidity at the moisture source, for example, changes occurring during vapor formation or moisture recycling processes (Landais et al., 2008; Luz and Barkan, 2010). This makes $\Delta^{17}O$ a powerful complementary tracer for detecting subtle changes in atmospheric circulation and regional hydrological processes.

Despite their importance, long-term records of precipitation isotopes over the Korean Peninsula remain limited. The Global Network for Isotopes in Precipitation (GNIP), established in the 1960s, provides baseline data at a global scale; however, its coverage in East Asia is sparse and often discontinuous (Aggarwal et al., 2010). This lack of high-resolution, continuous isotope data limits the understanding of hydrological processes in the Korean Peninsula and their response to climate variability and change. To address this, in this study, a highly temporal-resolution, multi-year dataset of ground-based stable hydrogen
and oxygen isotopes in precipitation was obtained over the Korean Peninsula to investigate their seasonal and interannual variability in this mid-latitude setting. The results of this study provide a foundation for investigating moisture source dynamics, the behavior of isotope tracers such as d-excess and $\Delta^{17}O$, and the long-term isotopic response to climate variability in East Asia. These data are also essential for evaluating isotope-enabled climate models and interpreting regional paleoclimate proxies, and could accordingly enhance the understanding of hydroclimatic processes in monsoon-affected regions. By employing
high-resolution, ground-based isotope observations, this study provides a critical step toward refining the interpretation of atmospheric processes in mid-latitude monsoon affected regions.



## 2 Study area and methods

The observations were made approximately 30 km inland from the western coast of the Korean Peninsula, on the campus of Ewha Womans University in Seoul, South Korea (37°33'53" N, 126°56'46" E; Fig. 1 (A) and (B)). The sampling point was located 23 m above sea level. The study area experiences a temperate monsoon climate, with large seasonal variations characterized by hot, humid summers driven by the East Asian summer monsoon (EASM) and cold, dry winters dominated by the Siberian High and the East Asian winter monsoon (Kim et al., 2019; Lee et al., 2013). In summer (June–August; JJA), moist southwesterlies bring significant rainfall (known variously as the *Changma* in Korea, *Meiyu* in China, and *Baiu* in Japan), leading to the concentration of precipitation. Conversely, in winter the extensive Siberian High strengthens, pushing cold, dry air southward across the Korea Peninsula (Ding and Chan, 2005). These contrasting patterns result in sharp seasonal transitions and frequent extreme weather events, such as torrential rainfall during the monsoon season and cold snaps accompanied by strong winds in winter (December–February; DJF). The study area experiences four distinct seasons: spring (March–May; MAM), summer, autumn (September–November; SON), and winter (Ha et al., 2012; Lee et al., 2013).

Precipitation samples were collected between January 2016 and December 2020 (five years) at approximately biweekly intervals. Sampling was timed to coincide with rainfall events and followed the method recommended by the Global Network of Isotopes in Precipitation (GNIP). Samples were collected on an unobstructed rooftop of a five-story building. This point was selected because its open setting, free from nearby buildings or vegetation, ensured that the collected precipitation was representative and minimally affected by local interference (Fig. 1C). To minimize contamination and evaporation, precipitation was collected using a funnel system before being immediately filtered through a 0.45 µm syringe filter (Membrane Filter, HYUNDAI MICRO Co., Seoul, South Korea) and transferred to pre-cleaned PTFE bottles. The samples were then stored at –20 °C until analysis. When snow fell, samples were first melted at room temperature while remaining sealed in their original HDPE bottle before being processed using the aforementioned protocol.

All samples were transported in a frozen state to the Korea Polar Research Institute (KOPRI), where water isotope analysis was conducted using a wavelength-scanned cavity ring-down spectrometer (WS-CRDS; model L2140-I, Picarro Inc., CA, USA). The analytical protocol followed the optimized method of Kim et al. (2022) for high-precision triple oxygen isotope measurements. This method includes a systematic approach to minimize memory effects, determine injection numbers, and calibrate the results using international reference waters. Each sample was injected 20 times per vial to minimize memory effects. Samples and reference materials were prepared in duplicate vials; only the second vial was used for evaluation, while the first served as a buffer against carryover effects. Of the total 20 injections per vial, only the final six were used to calculate the average δ-values, in order to minimize memory effects from preceding samples. This strategy significantly reduced the memory effects associated with isotopic differences between successive samples. Three international reference waters, Vienna Standard Mean Ocean Water (VSMOW), Standard Light Antarctic Precipitation 2 (SLAP2), Greenland Ice Sheet Precipitation (GISP) and one in-house standard were used for Vienna Standard Mean Ocean Water (VSMOW)-SLAP scale normalization.



To further mitigate memory effects, an intermediate-composition standard, GISP was measured between VSMOW2 and

SLAP2. Analytical precision was determined based on the long-term 1σ standard deviation of repeated measurements of international standards, accumulated over several years of laboratory operation. The corresponding precisions were 0.1, 0.07, and 0.01‰ for $\delta^2$H, $\delta^{18}$O, and $\delta^{17}$O, respectively. Every 10 samples (i.e., every 10 vials), a laboratory standard was performed to confirm the analytical conditions. All data were reported relative to VSMOW using the delta notation (δ) (Eq. (1)):

$$\delta(‰) = \left(\frac{R_{sample}}{R_{VSMOW}} - 1\right) \times 1000, \tag{1}$$

where $R_{sample}$ and $R_{VSMOW}$ represent the isotopic ratios of the sample (i.e., $^{18}O/^{16}O$, $^{17}O/^{16}O$, or $^2H/^1H$) and VSMOW, respectively.

To account for multiple precipitation events within a month, precipitation-weighted monthly means ($\delta_{wm}$) were calculated for $\delta^2$H, $\delta^{18}$O, and $\delta^{17}$O as follows (Eq. (2)):

$$\delta_{wm} = \frac{\sum P_i \times \delta_i}{\sum P_i} \tag{2}$$

where $P_i$ is the precipitation amount and $\delta_i$ is the isotopic value for each event.

## 3 Results

### 3.1 Variations in precipitation stable isotopes

A total of 130 precipitation samples were collected during the study period. The measured isotopic compositions of precipitation varied considerably: $\delta^{17}$O ranged from 0.61 to –9.62‰ (average: –3.75‰); $\delta^{18}$O from 1.15 to –18.21‰ (average:

–7.11‰); and $\delta^2$H from 3.3 to –132.0‰ (average: –46.6‰). The d-excess fluctuated between 24.9 and –5.9‰ (average: 10.4‰), whereas $^{17}O$-excess ranged from 69 to –28‰ (average: 16.8‰). Fig. 3 presents monthly box plots for these parameters with a sine-function fit. For all three parameters ($\delta^{17}$O, $\delta^{18}$O and $\delta^2$H), the precipitation isotopic values were relatively depleted during the coldest months (December to February), increased between around March and April as condition warmed, and then sharply decreased between June and August, when precipitation peaked. This pattern indicates a strong interplay between

temperature effects and rainfall intensity to describe the isotope signals. The monthly isotopic patterns observed during the study period align with those reported for Jeju Island in the Korean Peninsula and the Yangtze River region in China (Gou et al., 2022; Shin et al., 2021), suggesting that these regions can be influenced by similar meteorological patterns over East Asia. A distinct seasonal variation in d-excess is evident from the box plots (Fig. 3), with higher values in winter (particularly January and February; whole winter median values of 15–20‰) and lower values in summer (medians: 0–5‰). This pattern is

consistent with previous findings in the Korean Peninsula and reflects the sensitivity of d-excess to relative humidity and non-





equilibrium fractionation processes driven by moisture-source conditions (Kim et al., 2019; Lee and Kim, 2007). In winter, cold, dry air and a steep temperature/humidity gradient between the atmosphere and the ocean (or other moisture sources) enhance evaporation-driven fractionation, thereby elevating d-excess. Conversely, in summer, high relative humidity and abundant precipitation suppress these fractionation effects, resulting in substantially lower d-excess. Unlike d-excess, the [17]O-

excess was generally high in winter and early spring and then decreased in summer, when the median occasionally became negative (as shown by the black-dot outliers in Fig. 3). It subsequently rose again in autumn, revealing a clear seasonal pattern of high [17]O-excess during the cold and/or dry seasons (spring, autumn, and winter) and lower values during the warm and humid summer period. The increase in [17]O-excess during winter and early spring suggests that water vapor sourced from drier air masses (e.g., continental or high-latitude oceanic regions) undergoes more pronounced kinetic fractionation during this

period. Overall, the sine fit of the average monthly [17]O-excess values ($R^2 = 0.53$) supports the above interpretation; although not perfectly periodic, it exhibits a distinct seasonal cycle, peaking in winter to early spring and reaching its minimum in summer.

**3.2 Local Meteoric Water Line**

The relationship between the precipitation $\delta^{18}O$ and $\delta^2H$ defines a Local Meteoric Water Line (LMWL) that closely aligns

with the Global Meteoric Water Line (GMWL; Craig, 1961), while exhibiting additional seasonal variations (Fig. 4(A)). The LMWL derived from linear regression is $\delta^2H = 7.95 \cdot \delta^{18}O + 10.0$ ($R^2 = 0.98$), indicating that the isotopic compositions of precipitation in Seoul are primarily governed by equilibrium fractionation. The nearly identical slope to the GMWL suggests minimal deviation from global meteoric trends, although seasonal changes in moisture sources and humidity can introduce modest variation. Further examination of seasonal subsets revealed distinct differences in isotopic behavior across the year.

Summer precipitation clusters tightly along the GMWL, indicating nearly equilibrium condensation under humid monsoonal conditions. This pattern is consistent with the dominance of moisture, originating from the Northwest Pacific and South China Sea, where high humidity minimizes kinetic fractionation effects. Conversely, winter precipitation plots above the GMWL, with a higher intercept (~22) compared to summer, reflecting the influence of cold, dry air masses and enhanced d-excess.

This difference between summer and winter precipitation suggests that isotopic kinetic fractionation, likely associated

with ice-phase microphysics and dry air mass transport from the Asian continent, plays a greater role in winter precipitation compared to summer (Kim et al., 2019; Merlivat and Jouzel, 1979; Uemura et al., 2008). Spring and autumn values fall between these seasonal extremes, maintaining an overall LMWL slope close to 8. The persistence of a near-global slope across seasons indicates that equilibrium fractionation dominates the isotopic system; however, modest seasonal variations in intercept reflect differences in humidity and the moisture source over the year. These findings align with previous studies conducted in Korea

based on year-long precipitation isotope records from Jeju, Hongseung and Busan, which reported LMWL slopes, ranging from 7.3 to 8.4 and intercepts from 11.3 to 19.2 (Lee et al., 2003; Shin et al., 2021; Yoon and Koh, 2021). Compared to these, the LMWL in this study (slope = 7.95, intercept = 10.0) shows a similar slope but a slightly lower intercept. This confirms



that, while Korean precipitation follows global meteoric trends, seasonal shifts in air mass origin and fractionation processes introduce predictable deviations from these trends (Lee et al., 2007; Lim et al., 2012; Shin et al., 2021; Yoon and Koh, 2021).

The correlation between $\delta^{17}O$ and $\delta^{18}O$ in natural waters has been well-established to follow a nearly linear relationship under equilibrium conditions (Angert et al., 2004; Landais et al., 2008; Luz and Barkan, 2011). This relationship is a fundamental characteristic of stable oxygen isotopes in precipitation, with minor deviations due to kinetic effects, ice-phase processes, and variations in relative humidity at the moisture source (Barkan and Luz, 2005). In the present study, the relationship between precipitation $\delta^{17}O$ and $\delta^{18}O$ defines an LMWL, exhibiting strong linearity across all samples, although

including seasonal variability (Fig. 4(B)). A linear regression applied to the full dataset results in $\delta^{17}O = 0.528 \cdot \delta^{18}O + 0.0105$ ($R^2 = 1.00$), confirming the strong linear correlation between $\delta^{17}O$ and $\delta^{18}O$, indicative of characteristic of mass-dependent fractionation in meteoric waters. A distinct separation in slope occurs when precipitation is classified into winter and non-winter periods: The higher $\delta^{17}O$ and $\delta^{18}O$ slopes in winter precipitation reflect enhanced kinetic fractionation under cold and dry conditions, where low humidity amplifies non-equilibrium effects during condensation (Luz and Barkan, 2011). In this

season, Rayleigh distillation along moisture transport pathways further depletes heavy isotopes in precipitation, increasing the $\delta^{17}O$ and $\delta^{18}O$ slope, a pattern also observed in high-latitude precipitation (Landais et al., 2012). Ice-phase microphysical processes, particularly supersaturation with respect to ice, cause additional fractionation, reinforcing the seasonal difference in the slope of $\delta^{17}O$ vs $\delta^{18}O$ regression line (Luz and Barkan, 2010). In contrast, non-winter precipitation follows near-equilibrium fractionation, with high humidity minimizing kinetic effects and maintaining $\delta^{17}O$–$\delta^{18}O$ ratios similar to those of

global meteoric waters (Angert et al., 2004; Landais et al., 2008). These seasonal variations highlight the role of atmospheric humidity and cloud microphysics in modulating triple oxygen isotope fractionation.

## 4 Discussion

### 4.1 Climatic controls on precipitation isotope composition

We performed a seasonal correlation analysis between precipitation isotopes ($\delta^2H$, $\delta^{18}O$, d-excess, and $^{17}O$-excess) and

meteorological parameters (air temperature, relative humidity and precipitation amount) (Fig. 5). The overall monthly averages across the study period revealed a significant correlation between meteorological variables and isotopes only for d-excess. This suggests that variations in $\delta^{18}O$ and $^{17}O$-excess are governed by different meteorological influences that vary with season. In contrast, d-excess consistently exhibited significant negative correlations with relative humidity, temperature, and precipitation. These relationships can be attributed to a combination of factors: lower relative humidity and temperature at the

moisture source enhance kinetic fractionation during evaporation, thereby increasing d-excess (Merlivat and Jouzel, 1979; Uemura et al., 2008), while locally, higher temperatures and humidity may promote re-evaporation and mixing, which reduce d-excess (Steen-Larsen et al., 2014). The negative correlation with precipitation may reflect the amount effect, but is better interpreted as a result of multiple interacting meteorological factors (Holmes et al., 2024).





In spring, strong positive correlations were observed between $\delta^{18}O$ and both temperature and precipitation (r = 0.49 and 0.53, respectively; n = 10 in each case). This indicates a significant temperature effect during spring due to relatively dry conditions and intermittent precipitation. The d-excess also displayed strong negative correlations with temperature during this period, providing insights into moisture sources and isotopic fractionation during precipitation (Uemura et al., 2008). During summer, $\delta^{18}O$ was significantly negatively correlated with precipitation amount (r = –0.44, n = 14), reflecting the amount-effect that is typical in monsoon climates (Lee et al., 2015). Increased relative humidity during summer resulted in more frequent rainfall events, contributing to lower $\delta^{18}O$ values. Furthermore, the relatively weak correlation between d-excess and temperature observed during summer can be attributed to prolonged monsoon precipitation and the marine origin of air masses in this season. In autumn, the monthly mean $\delta^{18}O$ was strongly negatively correlated with precipitation amount (r = –0.55, n = 10), mainly due to frequent heavy rainfall events associated with typhoons or tropical cyclones, which are most common during this season. Although there were no direct passages of typhoons at the study site during the study period, substantial precipitation events influenced by typhoons were frequently observed. The strong correlations observed between d-excess and meteorological variables likely reflect moisture supply from nearby oceanic areas influenced by migratory high-pressure systems. In winter, d-excess and $\Delta^{17}O$ exhibited a clear negative correlation with meteorological variables such as temperature and precipitation amount, strongly reflecting evaporation conditions and the characteristics of moisture sources from the nearby ocean.

The results of this study indicate that seasonal variations in precipitation isotopes in Korea are closely linked to local meteorological factors such as temperature, relative humidity, and precipitation amount and reflect distinct seasonal regimes shaped by synoptic-scale circulation patterns. The findings indicate that, in summer, isotopic depletion is primarily governed by the amount effect under the influence of the East Asian monsoon, which delivers warm, moisture-rich air masses and produces intense rainfall events. Spring exhibits more variable isotopic signals due to transitional moisture sources and fluctuating atmospheric conditions, which result in a combination of continental and maritime influences. In autumn, isotopic variability is often enhanced by episodic typhoons, which introduce large volumes of isotopically light precipitation associated with strong convective activity. In contrast, winter precipitation is strongly depleted in heavy isotopes and enriched in d-excess due to the presence of cold, dry continental air masses advected by the East Asian winter monsoon. This seasonal variation underscores the role of changing moisture origins and precipitation mechanisms in modulating the stable isotope composition of precipitation across Northeast Asia, including the Korean Peninsula.

## 4.2 Interpreting seasonal decoupling of $\Delta^{17}O$ and d-excess

Variations in the $\Delta^{17}O$ and d-excess of meteoric water are primarily governed by kinetic fractionation, making them reliable indicators of relative humidity at the moisture source (Barkan and Luz, 2007; Landais et al., 2010; Pfahl and Sodemann, 2014; Uemura et al., 2008; Nyamgerel et al., 2021). However, when measured in precipitation, these isotopic indices may also reflect complex post-evaporation processes such as continental recycling, partial re-evaporation within clouds, and sub-cloud raindrop





evaporation (Landais et al., 2010; Li et al., 2015; Tian et al., 2018; Xia et al., 2023). These additional factors complicate the interpretation of seasonal isotopic variability in precipitation (Aron et al., 2023; Chen et al., 2023).

Our analysis showed that, during spring, summer, and autumn, $\Delta^{17}O$ was moderately negatively correlation with $\delta^{18}O$ and weakly positively correlated with d-excess (Fig. 6). These tendencies are broadly consistent with theoretical expectations
under non-steady-state evaporation, where kinetic fractionation induces a simultaneous increase in $\Delta^{17}O$ and d-excess and a depletion in $\delta^{18}O$ (Li et al., 2015). The slopes observed between $\Delta^{17}O$ and d-excess fall within the range of 0.7–2.0 per meg per ‰, which aligns with results from conceptual models and field-based estimates in regions influenced by oceanic moisture (Landais et al., 2010; Li et al., 2015). These observations suggest that, in non-winter seasons, kinetic processes such as evaporation and sub-cloud re-evaporation exert a dominant influence on isotopic composition. In contrast, in winter
precipitation, no statistically significant correlation was observed between $\Delta^{17}O$ and either $\delta^{18}O$ or d-excess. While the d-excess range remained relatively narrow in winter, $\Delta^{17}O$ values showed a larger dispersion in this season (Fig. 3). This may be attributed to the higher sensitivity of $\Delta^{17}O$ to water vapor mixing compared to d-excess, which is especially relevant under colder conditions and weaker surface moisture recycling (Xia et al., 2023). Due to its logarithmic formulation, $\Delta^{17}O$ is more susceptible to non-linear averaging during mixing than d-excess, and may therefore decouple from $\delta^{18}O$-based processes in
winter (Li et al., 2015; Xia et al., 2023). This suggests that, in winter, isotopic variability in precipitation is more likely to be influenced by mid-tropospheric mixing or contributions from diverse vapor sources than by surface evaporation conditions alone.

The results further indicate that $\Delta^{17}O$ is negative correlated with $\delta^{18}O$ but positively correlated with d-excess during spring, summer, and autumn, consistent with theoretical expectations under non-steady-state evaporation conditions. These
correlations reflect the influence of kinetic fractionation processes such as evaporation and sub-cloud re-evaporation, with $\Delta^{17}O$–d-excess slopes (0.7–2.0 per meg per ‰) aligning with previous modeling and observational studies (Landais et al., 2010; Li et al., 2015). In contrast, winter precipitation showed no significant correlations between $\Delta^{17}O$, $\delta^{18}O$, and d-excess, and moreover, showed increased $\Delta^{17}O$ variability compared to the other seasons, suggesting a greater sensitivity to vapor mixing and reduced surface recycling. Overall, these findings demonstrate the utility of $\Delta^{17}O$, alongside $\delta^{18}O$ and d-excess, in
disentangling the effects of evaporation, recycling, and mixing on precipitation isotopes. The distinct behavior of wintertime $\Delta^{17}O$ observed in this study highlights its potential as a diagnostic tool for tracing seasonal vapor transport and source-region shifts over East Asia, including episodic moisture influxes from the Yellow Sea.

### 4.3 Seasonal Comparison of Precipitation Isotope Trends: Observations vs. Iso-GSM Model

Fig. 7 compares the monthly mean stable isotope values ($\delta^{2}H$ and $\delta^{18}O$) and d-excess in precipitation from the observational
data obtained in this study (black line/gray shading) and those from the Iso-GSM (Isotope-enabled Global Spectral Model, Yoshimura et al., 2008) outputs (red line/pink shading). The Iso-GSM is a general circulation model that includes the transport





and fractionation of stable water isotopes, enabling the simulation of isotope signals throughout the hydrological cycle (Yoshimura et al., 2008). It has been widely applied in studies across various climatic regimes, including East Asia, and is particularly suited for analyzing seasonal variations in precipitation and temperature. In this study, we chose Iso-GSM due to

its established performance in reproducing isotope climatology at mid-latitudes (Bong et al., 2024; Kim et al., 2019; Yoshimura et al., 2008). While both datasets exhibit broadly similar seasonal trends, several systematic differences are evident. First, for $\delta^2$H and $\delta^{18}$O, the Iso-GSM values are more negative than the observations, particularly during late autumn and winter. This may reflect model biases in the representation of Rayleigh distillation during long-range vapor transport or limitations in resolving moisture source conditions under cold-season dynamics. Second, the modeled d-excess values show a relatively

narrow monthly range compared to observations; the observed data reveal large intra-annual variability in d-excess (ranging from ~5 to ~20‰), with variability except during winter season (DJF). In contrast, the Iso-GSM simulations simulate smaller variability than observed, showing consistently reduced deviations from the monthly means across the year. This likely arises from the model's limited representation of kinetic fractionation processes, particularly those linked to sub-cloud evaporation, partial raindrop re-evaporation, and moisture recycling. These limitations are consistent with findings from previous studies

highlighting the challenges in capturing d-excess variability using isotope-enabled General Circulation Models (Pfahl and Sodemann, 2014; Risi et al., 2008). Third, although some months showed overlapping uncertainty envelopes between the Iso-GSM simulations and the observations (i.e., ±1σ), the largest deviations were observed in the d-excess values, which indicates statistically significant mismatches between the model and observations. These divergences may be further exacerbated by simplifications in the cloud microphysics or sea surface temperature boundary inputs used in the model.

290           Taken together, the results suggest that, while the Iso-GSM model captures the broad seasonal pattern of $\delta^{18}$O and $\delta^2$H reasonably well, it underrepresents the amplitude and variability of d-excess, an index sensitive to kinetic fractionation and near-surface atmospheric processes. Model fidelity could be enhanced by improving the parameterization of cloud–precipitation interactions, moisture source tracking, and convective dynamics. Moreover, integrating additional observational constraints, for example, $\Delta^{17}$O, which is more sensitive to mixing and kinetic processes compared to $\delta^{18}$O and $\delta^2$H—may

provide complementary information for validating and refining isotope-enabled models. Future work could also use trajectory-based diagnostics to isolate the meteorological histories of vapor parcels to better constrain the causes of observed model–data mismatches.

## 5 Data availability

The data that support the findings of this work are openly available at PANGAEA:

https://doi.pangaea.de/10.1594/PANGAEA.983390 (S. Kim et al., 2025).





## 6 Summary

This study examined 130 precipitation samples collected in Seoul between 2016 and 2020 to quantify seasonal variability in $\delta^2$H, $\delta^{18}$O, $\delta^{17}$O, d-excess, and $\Delta^{17}$O and clarify how these isotopic tracers respond to local meteorological conditions. $\delta^{17}$O, $\delta^{18}$O, and $\delta^2$H followed a pronounced sinusoidal seasonal cycle, being most depleted during winter, becoming gradually

enriched in spring, and sharply declining during the summer monsoon due to the amount effect. d-excess was highest in winter and lowest in summer, reflecting its sensitivity to non-equilibrium evaporation and relative humidity at the moisture source. Meanwhile, $\Delta^{17}$O showed a similar seasonal cycle, although with reduced amplitude, highlighting its additional sensitivity to large-scale circulation and vapor transport. The calculated Local Meteoric Water Line ($\delta^2$H = 7.95·$\delta^{18}$O + 11.2) closely resembles the Global Meteoric Water Line, but with a higher winter intercept, suggesting enhanced ice-phase fractionation

and the influence of dry continental air masses. The $\delta^{17}$O–$\delta^{18}$O relationship confirmed mass-dependent fractionation across the dataset; however, the slope steepened during winter, indicating stronger kinetic effects under low humidity.

The comparison of the seasonal behavior of $\delta^{18}$O, d-excess, and $\Delta^{17}$O revealed distinct tracer-specific responses. During spring, summer, and autumn, $\Delta^{17}$O was negatively correlated with $\delta^{18}$O and positively correlated with d-excess, consistent with theoretical expectations for non-steady-state evaporation. The slope between $\Delta^{17}$O and d-excess ranged from 0.7 to 2.0

per meg per ‰, aligning with conceptual models and empirical results from ocean-influenced regions. In contrast, in winter, no statistically significant correlation was observed between $\Delta^{17}$O and either $\delta^{17}$O or d-excess, while $\Delta^{17}$O displayed greater dispersion to compared to the other season. This decoupling likely reflects the heightened sensitivity of $\Delta^{17}$O to mid-tropospheric vapor mixing and contributions from diverse moisture sources, rather than surface evaporation alone. These findings underscore the utility of $\Delta^{17}$O as a diagnostic tracer of atmospheric mixing and moisture transport, especially in cold

seasons.

By integrating $\delta^{18}$O, d-excess, and $\Delta^{17}$O, this study provides a more comprehensive understanding of seasonal hydrological processes than would be possible using any of these tracers alone. The results highlight key controls on isotopic variability in the East Asian monsoon system, particularly during winter, when interactions between continental and marine air masses become dominant. This dataset serves as a valuable benchmark for interpreting modern hydroclimatic dynamics

and offers a foundation for evaluating isotope-enabled climate models. Beyond contemporary climate diagnostics, the integrated use of $\Delta^{17}$O and d-excess also holds implications for interpreting isotope records in subsurface hydrological archives. These tracers may enable enhanced reconstructions of past climatic conditions from speleothems or glacier ice and improve the understanding of groundwater recharge processes in monsoon-influenced regions. As such, this work bridges modern atmospheric processes with paleoclimate interpretations and supports future hydroclimate modeling and water resource

management across East Asia.



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




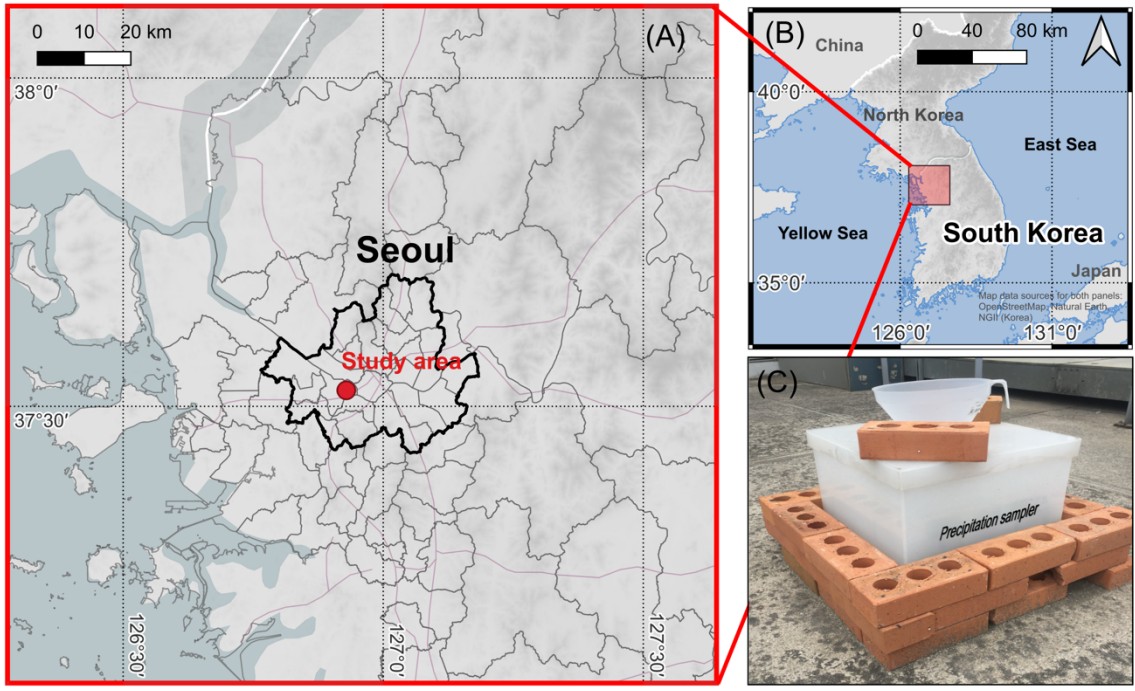

**Figure 1. (A)** Enlarged map showing the study site (red dot) in Seoul, South Korea, with administrative boundaries of surrounding regions. **(B)** Regional map placing South Korea within East Asia. **(C)** Precipitation sampling device installed at the study site, designed in accordance with GNIP (Global Network of Isotopes in Precipitation) guidelines to minimize post-collection evaporation. Maps in panels (A) and (B) were created by the authors using QGIS. Map data sources: © OpenStreetMap contributors (ODbL), Natural Earth, and the National Geographic Information Institute of Korea (NGII).



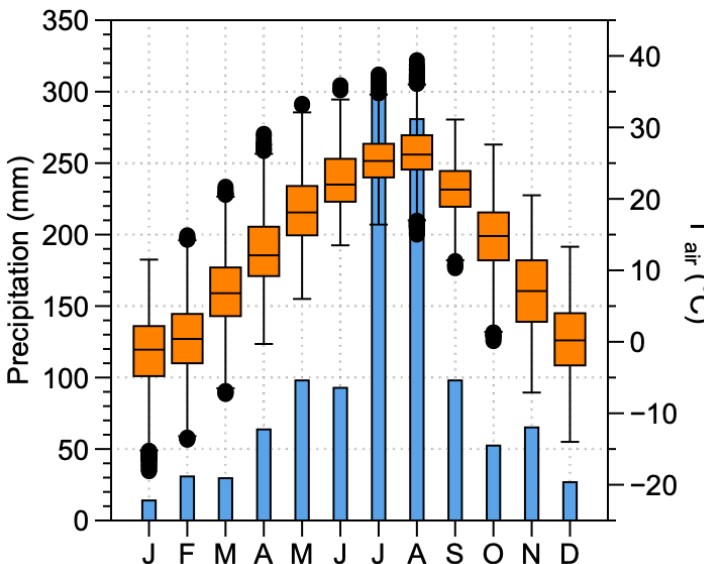


**Figure 2: The average monthly precipitation amount (grey bars) and average monthly temperature (black-lined boxes) for the city of Seoul, based on meteorological data provided by the Korea Meteorological Administration (available at: https://www.weather.go.kr/w/index.do).**

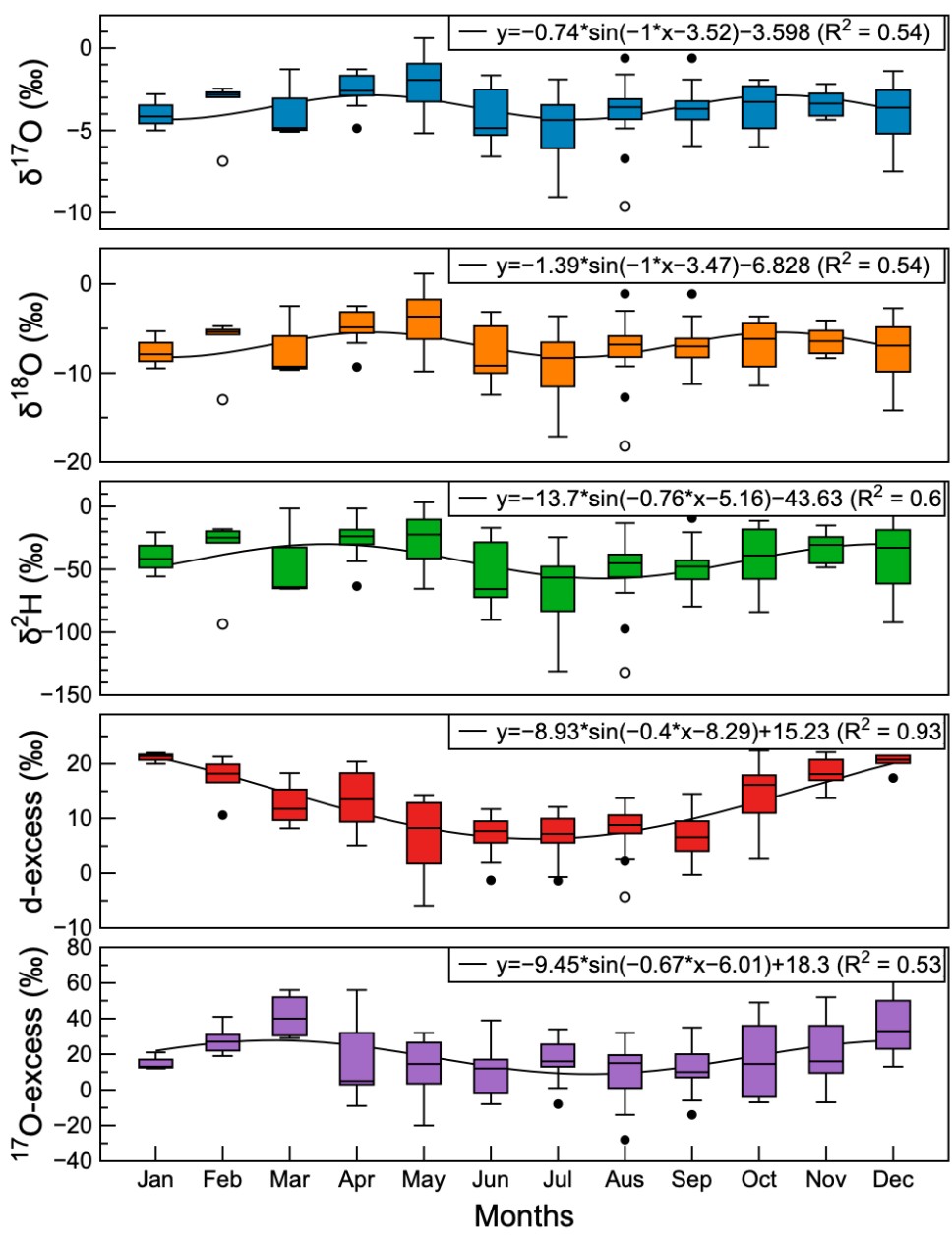

**Figure 3: The monthly average values of the observed isotopic tracers. From top to bottom: $\delta^{17}O$, $\delta^{18}O$, $\delta^2H$, deuterium excess (d-excess), and $^{17}O$-excess.**





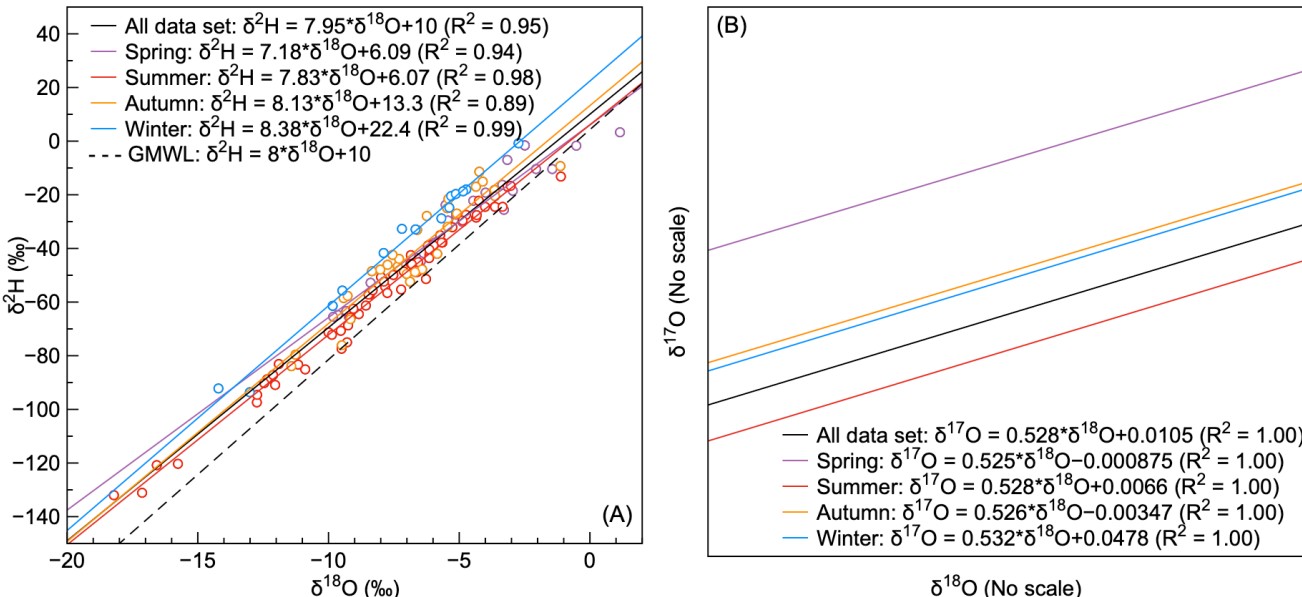

**Figure 4: The values of isotopic tracers by season, with regression lines for each. (A) The black regression line represents the aggregated data for the entire year, which served as a baseline for seasonal variations. Red points and their regression line indicate the summer trend. Blue points with the corresponding regression line depict the winter values. The purple line denotes the Global Meteoric Water Line (GMWL).**




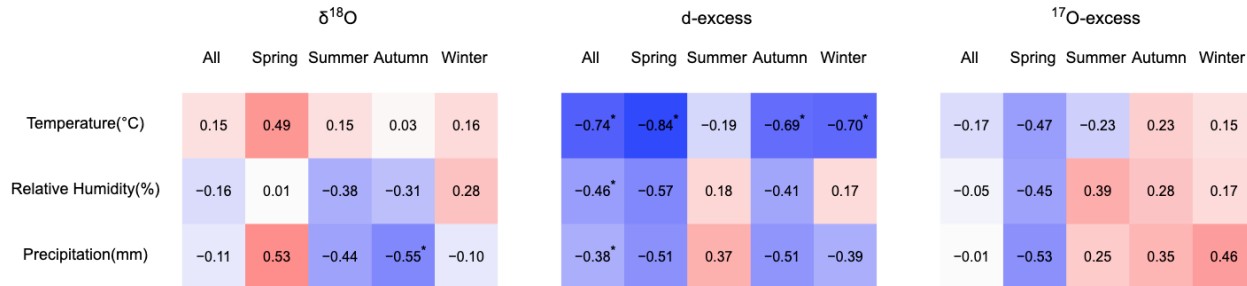

**Figure 5: Correlations between precipitation-weighted mean monthly precipitation isotopes ($\delta^{18}$O, d-excess, and $^{17}$O-excess) and the precipitation-weighted mean meteorological variables (air temperature, relative humidity) and total precipitation amount during the study period.**

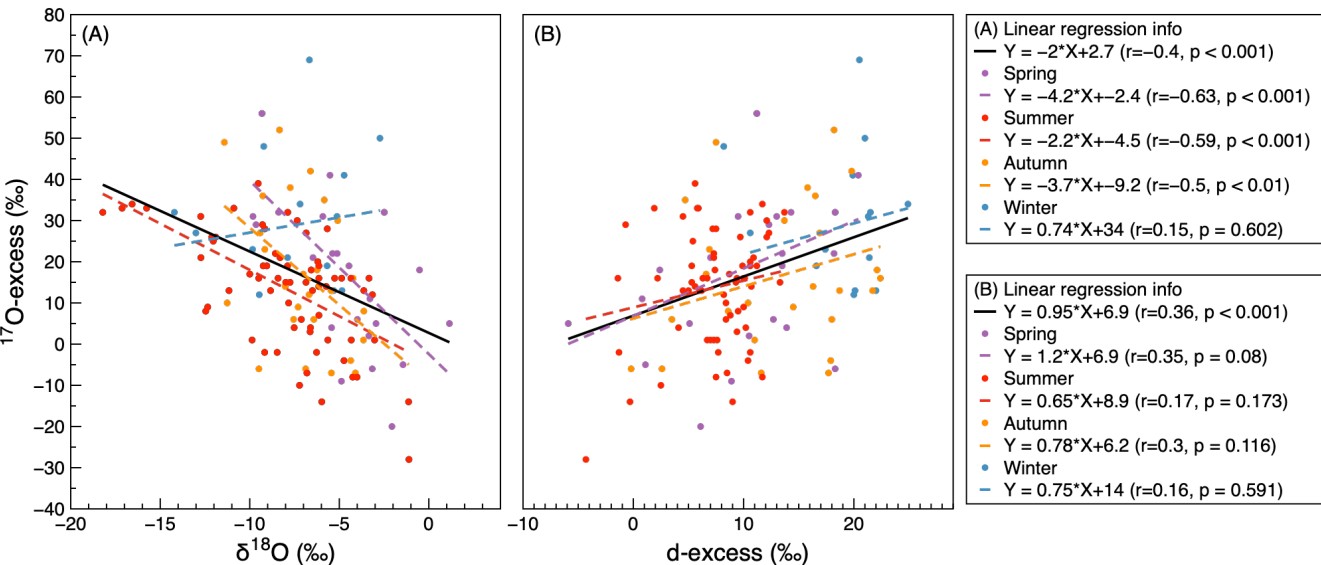

**Figure 6: The seasonal relationships between ¹⁷O-excess and (A) δ¹⁸O and (B) d-excess. Each marker represents a bi-weekly sample, color-coded by season (violet = spring; red = summer; orange = autumn; blue = winter). The regression line for each season is shown as a dashed line in the corresponding color, while the solid black lines represent the regression for the entire dataset. The regression equations and Pearson correlation coefficients (r) are listed in the legends.**


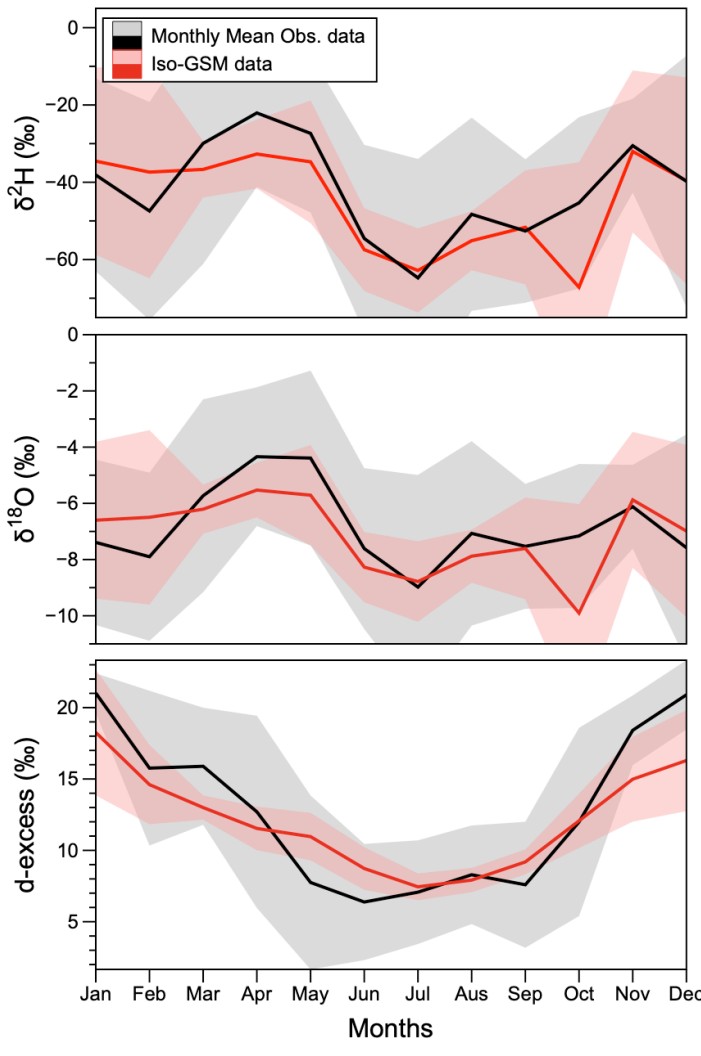

**Figure 7: The monthly mean precipitation stable isotope values (δ²H, δ¹⁸O, and d-excess) from observations (black solid line with gray shading) and Isotope-enabled Global Spectral Model (Iso-GSM) outputs (red solid line with pink shading). The shaded areas represent the uncertainty range of the monthly mean (±1 standard deviation), while the solid lines indicate the monthly average values.**
