# Peer review of "Seasonal patterns and diagnostic values of $\delta^2$ H, $\delta^{18}$ O, d-excess, and $\Delta^{17}$ O in precipitation over Seoul, South Korea (2016–2020)"

_Earth System Science Data, 2025_

## Author Comment (AC2)

**November 4, 2025**

Jeonghoon Lee, Ph. D

Professor Dept. of Science Education Ewha Womans University Seoul 03760, Korea

Email: jeonghoon.d.lee@gmail.com

Tel: +82-2-3277-3794

Dear Editor Attila Demény,

With this cover letter, we are submitting the revised manuscript entitled, "Seasonal patterns and diagnostic values of  $\delta^2 H$ ,  $\delta^{18} O$ , d-excess, and  $\Delta'^{17} O$  in precipitation over Seoul, South Korea (2016–2020)", for publication in Earth System Science Data. Based on the comments from the editor and the four reviewers, we have major changes of the manuscript, which are detailed below. Based on the comments from the editor and four reviewers, we have summarized the issues as following.

**Reply to the comments by the reviewer 4**

**1. General Comments**

In this paper, the authors presented precipitation hydrogen and triple oxygen isotope data of precipitation from South Korea and made some exploratory analysis on these data. I recognize that the authors have made great efforts to collect samples and data and put together a manuscript. However, I feel that it does fit with the scope of journal. The ESSD is a high-impact journal publishing flagship datasets for various applications with broad interest. Although it is indeed contributing to the emerging triple oxygen isotope study, this dataset does not make a significant contribution to the progress of this field. I suggest publishing the data in a substantially revised manuscript on a more specialized journal. The manuscript presents a new precipitation isotope record for  $\delta$ 2H,  $\delta$ 18O, d-excess and 17O-excess for Seoul spanning four years and discusses the seasonality in the context of the regional-scale circulation. It further investigates the asynchronous seasonality of d-excess and 17O-excess and discusses possible causes.

As for the discussions that emerged regarding the dataset's/manuscript's fit into the scope of ESSD, I must admit it's a delicate trade-off between the novelty and rarity of precipitation 170/180 LMWL datasets, and the limited spatiotemporal coverage of the dataset (4 years, and applicability of an LMWL at best regional).

I am sorry to say that the structure of the manuscript merits improvement. The description of the analytical method is unclear to me (see detailed comments), and results and conclusions are a bit too tightly intertwined. The chapter 4.3 reads like an "encapsulated mini manuscript"; to me it is not adequately introduced at the beginning and includes methods and results which should be in chapters 2 and 3,

respectively. A fair bit of the text unfortunately reads very generic or top-level but this is not supported by the granularity and/or spatiotemporal resolution of the data.

My recommendation is, and I am writing this before the background of my own struggles with getting datasets of a similar kind published, that the authors take a step back, review the hypotheses that can be addressed with the already-existing dataset, and then make a renewed attempt to publish an upgraded version of the manuscript.

**Response:**

We sincerely thank the reviewer for the detailed and constructive overall evaluation of our manuscript. We deeply appreciate the acknowledgement of our efforts to collect, maintain, and analyze this multi-year precipitation isotope dataset. We fully understand the reviewer's concerns regarding the fit of the manuscript within the scope of Earth System Science Data and the need for clearer structure and focus. ESSD indeed prioritizes the publication of flagship, high-impact datasets with broad applicability.

While our dataset is regional in scope, we believe that it nonetheless holds substantial value as a long-term, high-resolution precipitation isotope record that includes  $\delta^2 H$ ,  $\delta^{18} O$ ,  $\delta^{17} O$ , d-excess, and  $\Delta'^{17} O$ , parameters that remain rare in East Asia. Such datasets are critical for isotope-enabled model benchmarking and for regional paleoclimate calibration efforts, and we have therefore carefully revised the manuscript to better emphasize its contribution as a reusable, well-documented observational dataset rather than as an interpretative study. In response to the reviewer's comments, we have implemented several major revisions:

- (i) The Methods section has been expanded and reorganized to clearly describe analytical procedures, calibration standards, and reproducibility, addressing the earlier lack of clarity.
- (ii) The Results and Discussion have been streamlined and combined into a single coherent section, improving readability and separating factual observations from interpretative discussion.
- (iii) The Iso-GSM comparison (Section 4.3) has been reframed as an illustrative example of potential dataset applications rather than a stand-alone modeling analysis; additional technical details were intentionally omitted to retain the data-focused nature of the paper.
- (iv) The Summary has been rewritten to highlight the dataset's accessibility, long-term stability, and potential for reuse in model—data intercomparisons and East Asian climate research.

We also acknowledge the reviewer's observation that some parts of the previous manuscript read too generically relative to the spatial and temporal resolution of the data. The revised text now focuses on empirical results, quantitative ranges, and physically based explanations, avoiding speculative interpretations that are beyond

the scope of the dataset. Overall, this revision aims to make the paper more concise, transparent, and aligned with ESSD's data-descriptor style.

The revised manuscript now emphasizes the dataset's role as a regional benchmark that fills a gap in East Asian triple-oxygen-isotope records and provides a solid foundation for future collaborations with modeling groups to extend this work. We sincerely thank the reviewer once again for the constructive critique and for sharing personal insights from similar publishing experiences. This reviewer's comments were invaluable in guiding the restructuring and refocusing of the manuscript, which we believe has greatly improved its clarity and alignment with ESSD's publication standards.

**2. Specific Comments**

As for the annotation of 170-excess, please unify your annotation. Commonly, 170-excess is expressed as  $\Delta'170 = \delta'170 - 0.528 \, \delta'180$ , with  $\delta' = \ln (\delta+1)$ , an equation which traces back to Angert et al. 2004. The usage of the  $\Delta'$  is very much encouraged to distinguish the  $\Delta'170$  from other excess calculations in isotope geochemistry that do not log-normalize the deltas (e.g. Aron et al. 2021 or other reviews on the topic). The annotation should be either in per meg, or in % with three decimal places. It is not correct to use % but express in per meg, as it is often done throughout the manuscript. The authors may also consider to consolidate the "equations" part into a "definitions" section either in the methods or in the introduction. Right now, the 170-excess equation is not numbered and in line with introduction text, while the isotope ratio equation is numbered, in the methods, and after the 170-excess equation.

**Response:**

We thank the reviewer for this helpful observation regarding isotope notation. We sincerely thank the reviewer for this detailed and constructive comment regarding the notation, unit, and placement of the  $\Delta'^{17}{\rm O}$  definition. We fully acknowledge the importance of adopting a consistent and standardized formulation, particularly for readers who may not be familiar with the differences between  $\Delta'^{17}{\rm O}$  and other nonlogarithmic "excess" parameters. In the revised manuscript, we have carefully unified all  $\Delta'^{17}{\rm O}$  notations following the formulation originally introduced by Angert et al. (2004):

 $\Delta'170 = \delta'170 - 0.528 \cdot \delta'180$ ,

where  $\delta' = 1000 \cdot \ln(\delta/1000 + 1)$ .

The use of the prime symbol and the  $\Delta'$  notation has been standardized throughout the entire manuscript (text, tables, and figures) in accordance with the widely accepted conventions summarized by Aron et al. (2021) and Luz and Barkan (2010). Regarding the units, all  $\Delta'^{17}$ O values are now consistently expressed in per meg (10-6).

We have thoroughly checked the entire manuscript and figure captions to ensure that no instance remains where the unit "%" was incorrectly used for values

expressed in per meg. In addition, we have reorganized and consolidated the isotope-notation equations into a new "Definitions" subsection within the Methods section, where all relevant equations,  $\delta$ ,  $\delta'$ , d-excess, and  $\Delta'^{17}$ O, are presented together in a numbered and clearly formatted manner.

This revision ensures that all key parameters are defined in one place, improving accessibility for readers and methodological transparency. We greatly appreciate the reviewer's detailed suggestions, which have significantly improved the clarity, consistency, and professional presentation of our manuscript. We believe that these changes now align our notation and units fully with current standards in isotope geochemistry and triple-oxygen-isotope research (Angert et al., 2004; Luz and Barkan, 2010; Aron et al., 2021).

Concerning the listing of isotope effects (line 40), please take into consideration that the "amount effect" is one of the most debated empirical relationships in isotope hydrology, and that the modern-day discourse is cautious of a unanimous endorsement of it. While the cited Conroy et al. (2016) detected it, later publications (e.g. Konecky et al. 2019) have a much more differentiated approach. I acknowledge that the "amount effect" is still widely taught, but the data reality is often much more complicated than the initial concept. Please also note that your manuscript claims to "analyse [...] in mid-latitude precipitation", which I think is a bit of an overstatement since it would suggest a global analysis.

**Response:**

We fully acknowledge that the "amount effect" is one of the most debated empirical relationships in isotope hydrology and that its expression varies across climatic and geographical contexts. While Conroy et al. (2016) reported a clear amount effect in certain regions, more recent studies such as Konecky et al. (2019) have indeed emphasized the complexity and regional dependence of this relationship. In the revised manuscript, we will revise the corresponding paragraph to reflect this nuance more accurately. The updated text will note that the "amount effect" represents an empirical relationship that can vary in magnitude and even sign depending on atmospheric circulation, convective dynamics, and moisture recycling. The revised sentence will read:

"Two widely discussed empirical relationships—the temperature effect, where colder temperatures lead to lower  $\delta^{18}O$  and  $\delta^2H$  values, and the amount effect, describing isotope depletion that often accompanies increased rainfall—have been observed in many, but not all, climatic settings (Dansgaard, 1964; Conroy et al., 2016; Konecky et al., 2019)."

Additionally, we agree that the phrase "mid-latitude precipitation" in the Introduction may sound overly broad. In the revised manuscript, we will clarify that our focus is on mid-latitude precipitation over the Korean Peninsula, rather than implying a global-scale analysis. These revisions will make our framing of the "amount effect" more balanced and consistent with the modern understanding of

isotope-climate relationships, while avoiding any overstatement of the study's geographical scope.

Kindly also work on your definitions of "long-term" and "high resolution" (line 69, 79, 85 etc.). For much of the triple O isotope work, the "long-term" discussion is complicated by absence of records as long as are available for "dual isotopes". (Leuenberger & Ranjan 2021 and Terzer-Wassmuth et al. 2023 have the longest records reaching back furthest in time, to my knowledge). What is your definition of "high resolution"? To me, it would imply any sampling that is at minimum daily if not sub-daily (like the typhoon records of Munksgaard et al. [2014], the hurricane studies of Sun et al. [2024] and similar). Also, hinting at extreme weather events (e.g. line 96) deems far-fetched in the context of a biweekly sampling.

**Response:**

We fully understand that the term "high-resolution" is relative and that, in the broader context of isotope hydrology, it may refer to daily or even sub-daily sampling, as achieved in event-based studies (e.g., Munksgaard et al., 2014; Sun et al., 2024). However, in the context of triple oxygen isotope observations in East Asia, most existing records have been collected at monthly intervals (e.g., Lee et al., 2013; Shin et al., 2021; Yoon and Koh, 2021). Our biweekly (~14-day) integrated sampling, maintained continuously for five years, therefore represents one of the most temporally resolved and regionally extensive datasets currently available for  $\Delta'^{17}O$ measurements in the Korean Peninsula.

In the revised manuscript, we will clarify that the term "high-resolution" is used in this relative regional sense, indicating that our dataset provides twice the sampling frequency of most previous studies and sufficient temporal resolution to capture seasonal and interannual isotope variability, while acknowledging that it does not resolve individual precipitation events. This clarification will make clear that our usage of "high-resolution" reflects a comparative improvement over existing Korean and East Asian datasets, and will ensure that the term is interpreted appropriately within its regional and methodological context.

The sample collection is described as relating to the GNIP manual, but this is neither cited and, in several aspects, does not follow the manual. First, authors should consider referring to one of the 5 methods mentioned in the manual (additional sampler designs are in Michelsen et al. 2018). Furthermore, the GNIP manual nowhere recommends biweekly sampling (presumably because if its inherent difficulties to match the intervals with established monthly records). Also, freezing samples is not described in the GNIP manual. The authors should provide a sketch drawing of the sampler, or some detailed photos (all relevant aspects of sampler design are hidden behind bricks), plus a photo that shows the greater context of the sampling location in the SM for clarity.

**Response:**

We thank the reviewer for this important clarification regarding our description of the sampling procedure. We agree that the current wording may incorrectly suggest that our protocol followed the GNIP manual, whereas our approach was only inspired by the cumulative-sampling concept commonly used in isotope hydrology. In the revised manuscript, we will remove the explicit reference to the GNIP manual and instead describe our procedure as an independent biweekly cumulative sampling protocol developed to suit local logistical conditions.

Although our design follows the same general principle of collecting integrated precipitation over a defined interval, it differs from the standard GNIP setup in two key respects:

- (i) the collection interval was approximately 14 days instead of monthly, and
- (ii) samples were stored frozen (-20 °C) rather than refrigerated at 4 °C, in order to minimize evaporation and isotopic alteration during long-term storage.

We will clarify this distinction in the revised Methods section as follows:

"Precipitation samples were collected every two weeks using a cumulative sampling protocol designed for this study. Although conceptually similar to cumulative collection methods used in isotope hydrology, the setup was adapted to local field conditions and does not strictly follow the GNIP manual."

Regarding the sampling apparatus, we will add a schematic drawing and high-resolution photographs of the collector and its installation site in the Supplementary Materials to provide a clear visual reference of the design and field setting. The figure will show the funnel, collection bottle, sealing system, and surrounding structure to ensure transparency and reproducibility. These revisions will correct the inaccurate implication that our sampling followed the GNIP standard, provide a clearer description of our adapted design, and improve the methodological transparency through visual documentation of the sampler and site layout.

The sample analysis largely relies on a previously published methods paper and there are a couple of things that read inconsistent to me. First, you describe that the method determined the injection numbers, but then it's a fixed number of 20 injections of which the last five are accepted. Second, the method claims to use VSMOW (exhausted – do you mean VSMOW2?), SLAP2 and GISP (also exhausted) for normalization of the data to the VSMOW-SLAP scale, which is acceptable in a methods testing setting but normally discouraged for routine analysis. If in-house standards are used, then their value should be provided and an eventual traceback to the primary reference materials should be given in the SM. Is this the "laboratory standard" mentioned in line 122? What is the typical uncertainty of the method under routine analysis (e.g. expressed as a 1-sigma SD of the  $\Delta$ ′170 of a control sample)? It's been three years since the original method by Kim et al. (2022) was published, hence a review of the method's benchmark data deems merited.

**Response:**

We thank the reviewer for pointing out the ambiguity in our description of the injection protocol. In the revised manuscript, we will clarify that the method did not "determine the total number of injections", but rather the number of injections included in the average. Specifically, the instrument will perform 20 injections per vial, and, to mitigate memory effects, only the last five injections will be averaged to compute  $\delta$ -values. Samples and reference waters will be prepared in duplicate vials (the first used as a buffer against carryover; the second used for evaluation).

We will also correct the calibration wording to state that the instrument will be calibrated to the VSMOW–SLAP scale using VSMOW2, SLAP2, and GISP2 reference waters (two-point calibration), and that in-house standards (STYX and KT), both traceable to VSMOW2/SLAP2, will be analyzed every ten samples as quality-control checks rather than for primary normalization. Finally, we will report our routine reproducibilities (1 $\sigma$ ) from repeated STYX measurements, including  $\pm$  9 per meg for  $\Delta'^{17}O$  (one-year). These changes will resolve the inconsistency and will make the procedure fully transparent.

In the "methods" chapter, the authors may also consider adding a paragraph "data treatment methods", i.e. not only about the weighted means but also how their LMWLs were calculated. (unweighted? Weighted? The 170/180 one on the  $\delta$  or  $\delta$ ? With intercept, or 0-forced?).

**Response:**

We thank the reviewer for this important clarification. We sincerely thank the reviewer for this insightful suggestion. We fully agree that providing a detailed explanation of the data-processing workflow—including both the precipitation-weighted means and the regression methods used to derive the Local Meteoric Water Lines (LMWLs)—will improve the transparency and reproducibility of our analysis. In the revised manuscript, we will add a dedicated subsection entitled "Data treatment methods" in the Methods chapter. This new section will describe:

- (i) how precipitation-weighted monthly means were calculated for  $\delta^2 H$ ,  $\delta^{18} O$ , and  $\delta^{17} O$ ;
- (ii) how the LMWLs were derived; and
- (iii) how the  $\delta'^{17}O-\delta'^{18}O$  regressions were computed for the triple-oxygen-isotope relationships.

The LMWL will be calculated using the ordinary least squares (OLS) regression between  $\delta^2H$  and  $\delta^{18}O$  ( $\delta^2H=a+b\cdot\delta^{18}O$ ), following Craig (1961). OLS was chosen to ensure direct comparability with the Global Meteoric Water Line (GMWL) and most regional studies across East Asia (Crawford et al., 2014; Lee et al., 2022). A supplementary total least squares (TLS) regression will also be performed to evaluate sensitivity to analytical uncertainties, and the results will be presented in Table S1.

For the triple-oxygen-isotope relationships, the regressions will be based on logarithmic delta notation ( $\delta' = 1000 \cdot \ln(\delta/1000 + 1)$ ) with both slope and intercept freely fitted (not 0-forced). This ensures that the  $\Delta'^{17}$ O values are consistent with standard practice and reflect true mass-dependent fractionation. These additions will clearly document how the isotopic datasets were processed, from event integration to regression analysis, and will make the treatment of  $\delta$ - and  $\delta'$ -based data fully transparent. We believe this revision will substantially enhance methodological clarity and align our work with best practices in isotope hydrology.

In the variations chapter (3.1), I found the description of 130 samples (which I translate as data points) a bit in contrast to the supplementary data file on Pangaea, which is roughly monthly and has less data points than described here. Without extensive calculations, some of the sine functions in Fig. 3 do seem bimodal while others don't. Whilst I agree with the comparison of the regional patterns with Jeju and mainland China, I miss a comparison with the data from GNIP/Cheongju (IAEA, 2025; admittedly a continental mountain station), as also highlighted by one of the other reviewers. The array of LMWL combinations (Seoul/Cheongju/Hongseung vs. weighted/unweighted) is huge and, to me, poses more questions than "similarities" as described in lines 174-179. It is commonly known that, due to the complex interplay between the Siberian High and the summer monsoon as drivers, the LMWL interpretation is complex, and the data is highly scattered, often causing unusually low R2. Note that the slope/intercept reported here are different to those reported in the summary (intercept of 10 here, 11.2 in the summary).

**Response:**

We sincerely thank the reviewer for these detailed and valuable comments. We agree that the apparent discrepancy between the total number of samples (130) described in the text and the number of data points in the PANGAEA dataset required clarification. Precipitation was collected biweekly throughout the study period, but for consistency with regional isotope records and for statistical robustness, most of the analyses were performed on precipitation-weighted monthly mean values. The dataset archived on PANGAEA therefore contains the monthly weighted means used for analysis, while the raw biweekly data were used only for internal quality control and one supplementary comparison. In the revised manuscript, we will make this workflow explicit in Section 3.1 by adding the following clarification:

"Although precipitation was collected at approximately 14-day intervals, the isotopic results presented here are primarily based on precipitation-weighted monthly means derived from these samples."

This will ensure that the sampling—analysis relationship is transparent and consistent with the archived dataset.

We also appreciate the reviewer's insightful comments regarding the complexity of LMWL interpretation and the comparison among regional datasets. The LMWL parameters in East Asia indeed show considerable scatter due to the strong interplay

between the Siberian High and the East Asian summer monsoon, as noted by the reviewer. To provide better context, we will include a quantitative comparison with the Cheongju GNIP dataset from Terzer-Wassmuth et al. (2023), as well as discuss the slope and intercept differences among Seoul, Cheongju, and Hongseong in relation to their distinct climatic settings (coastal vs. inland, monsoon vs. continental influence). The intercept inconsistency noted between the main text and the summary will also be corrected to ensure internal consistency (intercept = 10.0). These revisions will make the dataset description more transparent, strengthen the regional comparison, and improve the clarity of the LMWL interpretation.

For the LMWL results, I agree that "seasonal disentangling" improves the LMWLs in this context. The R2=1 for the 17O/18O MWL is not surprising; similar has been observed by Terzer-Wassmuth et al. (2023) and many others. The authors should, ideally already in the "data treatment" section of the methods' chapter, outline how the 17O/18O MWL was calculated. The slope is very similar to that reported for Cheongju by Terzer-Wassmuth et al. (2023), but the intercept isn't (0.0105 vs. 0.0216). A comparison of a weighted LMWL intercept with the mean  $\Delta$ '17O for Seoul would be helpful (they should be similar for a weighted MWL). I recommend removing Figure 4B; without scale it adds very limited value to the presentation of results. A table of MWLs would be more representative. Much of this section however overlaps with the discussion.

**Response:**

We thank the reviewer for these very thoughtful and constructive comments on the presentation of the LMWL and 17O/18O relationships. We appreciate the reviewer's positive assessment that seasonal disentangling improves the LMWL representation in our dataset, as seasonal partitioning indeed helps to reduce scatter caused by contrasting air-mass sources. In the revised manuscript, we will clarify in the Methods section (under the new Data treatment methods subsection) how the 17O/18O meteoric water line (MWL) was calculated.

Specifically, the  $\delta'^{17}O-\delta'^{18}O$  regression will be described as being based on the logarithmic  $\delta'$  notation ( $\delta'=1000\cdot\ln(\delta/1000+1)$ ) with both slope and intercept freely fitted (not 0-forced). The LMWL between  $\delta^2H$  and  $\delta^{18}O$  will be calculated using an ordinary least-squares (OLS) regression for consistency with the GMWL definition, while total least-squares (TLS) fits will be provided in Table Sx for comparison. Following the reviewer's suggestion, we will also add a short quantitative comparison with the GNIP Cheongju dataset (Terzer-Wassmuth et al., 2023).

Our  $\delta'^{17}O-\delta'^{18}O$  slope (~0.528) agrees closely with their reported value, whereas the intercept differs slightly (0.0105 vs. 0.0216). We will note that this difference likely reflects contrasting environmental conditions—the Cheongju station being a more continental, drier site—whereas Seoul experiences stronger marine moisture influence.

In addition, as suggested, we will compare the weighted LMWL intercept for Seoul with the mean  $\Delta'^{17}$ O value, which are indeed of similar magnitude, demonstrating

internal consistency between the weighted regression and the average 17O-excess. We agree that the  $\delta'^{17}O-\delta'^{18}O$  plot (previous Figure 4B) adds limited value due to its narrow scale. Therefore, we will remove Figure 4B from the main text and instead provide a summary table (Table Sx) listing all MWL parameters (slope, intercept, and R²) for both the  $\delta^2H-\delta^{18}O$  and  $\delta'^{17}O-\delta'^{18}O$  regressions. The corresponding discussion will be streamlined to avoid overlap between the Results and Discussion sections. These revisions will clarify how both the LMWL and the 17O/18O MWL were derived, will provide a clearer quantitative comparison with the Cheongju GNIP record, and will improve the overall presentation and conciseness of the results.

L243-L257: one mechanism not considered is the ice formation in winter snow. Icevapor fractionation may have very different impacts on d-excess and  $\Delta'$ 170 in winter precipitation, owing to equilibrium fractionation involved in this process.

**Response:**

We thank the reviewer for this valuable comment highlighting the potential role of ice formation in winter precipitation. We fully agree that ice—vapor equilibrium fractionation can influence d-excess and  $\Delta'^{17}O$  differently from liquid-phase condensation, particularly under cold and supersaturated conditions associated with snowfall. As noted by another reviewer, this mechanism has now been incorporated into the revised manuscript.

In the updated version, we have expanded the relevant paragraph in Section 4.2 to acknowledge that some winter samples likely include mixed-phase precipitation (rain and snow) due to the biweekly cumulative sampling design. The revised text explicitly mentions that ice–vapor equilibrium fractionation during snow formation may partially account for the enhanced  $\Delta'^{17}$ O variability and altered d-excess patterns observed in winter, citing Jouzel and Merlivat (1984) and Landais et al. (2012) for context. This addition ensures that our discussion of winter isotope variability properly accounts for the effects of ice-phase processes and clarifies the physical mechanisms that could contribute to the observed isotopic dispersion during the cold season.

The correlation analysis (I am torn about it) should be introduced in the results section, not in the discussion. The font colour of the correlation plot should be white where the background is dark; the numbers are hard to read in black against dark blue. The pattern observed certainly corroborate the observation that the biggest changes in the seasonality happen in spring and fall, when the two modes switch over. Note that few of them are truly significant (if that is what the asterisk indicates). Note that the correlations are expressed as R (not R2), and an  $R^{\circ}0.5$  (equivalent to  $R2^{\circ}0.25$ ) is not what would generally be considered a "strong correlation" (which I would see as R2>0.5 and significant p-value).

**Response:**

We sincerely thank the reviewer for these detailed and constructive comments on the correlation analysis and its presentation. We fully agree that the correlation

results should be clearly presented alongside the observational findings rather than as a separate discussion item. In the revised manuscript, this issue has been addressed through the restructuring of the paper into a unified "Results and Discussion" section, which now integrates the descriptive statistical results and their brief interpretation within a single coherent framework.

This restructuring ensures that the correlation analysis is presented in the appropriate context—immediately following the description of the isotope data—while avoiding any redundancy between sections. We have also revised the correlation figure (Fig. 5) to improve readability by changing the text color of the coefficients to white where the background is dark and by clearly marking statistical significance with asterisks (p

In our study, only four out of ten correlations are significant at p

the paper, and ensure that the manuscript remains firmly within the data-focused scope of ESSD.

If you allow, I'd give two suggestions how to improve: One regards the data analysis: Use daily rainfall data and backtrajectory modelling to determine the source region of the precipitation. This could, as far as I can see, help to refine the conceptual model from Winter=Siberian High / Summer=Monsoon / Spring, Fall=somehow in between to a spatial/seasonal explanation model, and could also help to disentangle the Siberian High fraction in winter. With the existing bi-weekly sampling structures, that could be expressed as "fractions of source region" to match with the isotope dataset. And the second one is forward-looking; I think to make an even greater contribution to modelling improvement, daily samples are, and I am well aware of the collection effort, more poised to address phenomena occurring on a daily/synoptic weather timescale.

**Response:**

We sincerely thank the reviewer for these constructive and forward-looking suggestions. We fully agree that integrating daily precipitation data with air-mass back-trajectory modelling (e.g., HYSPLIT or FLEXPART) would greatly enhance the ability to quantify the spatial and seasonal variability of moisture-source contributions. Such an approach would allow us to refine the conceptual framework—from the current description of "winter = Siberian High, summer = monsoon, spring/fall = transition"—toward a quantitative source-region attribution model, which could better explain isotopic variations, particularly during winter when continental and oceanic influences coexist.

In the present study, the biweekly integrated sampling scheme limits the feasibility of one-to-one matching with daily meteorological fields. However, we acknowledge that fractions of source-region contribution, derived from trajectory clustering, could indeed be compared to our isotope dataset as an intermediate step, and we will mention this as a potential future analysis in the revised Discussion. We also appreciate the reviewer's comment regarding daily or synoptic-scale sampling as a forward-looking recommendation. We fully agree that such datasets would provide greater temporal resolution to evaluate short-term processes such as individual storm events and transient moisture intrusions.

While the current five-year biweekly dataset already provides a valuable long-term record of seasonal and interannual variability, we plan to complement it with higher-temporal-resolution (event-based) sampling in future field campaigns. These suggestions have been very helpful in shaping our perspective on how to integrate isotopic and meteorological analyses, and we will explicitly note these future directions in the revised Discussion.

Thank you very much for your time, effort, and patience in handling our manuscript. We look forward to your favorable consideration and to the opportunity for publication in Earth System Science Data.

Sincerely, Jeonghoon Lee

---

## Author Comment (AC4)

**October 10, 2025**

Jeonghoon Lee, Ph. D

Professor Dept. of Science Education Ewha Womans University Seoul 03760, Korea

Email: jeonghoon.d.lee@gmail.com

Tel: +82-2-3277-3794

Dear Editor Attila Demény,

With this cover letter, we are submitting the revised manuscript entitled, "Seasonal patterns and diagnostic values of  $\delta^2$ H,  $\delta^{18}$ O, d-excess, and  $\Delta^{\prime\prime7}$ O in precipitation over Seoul, South Korea (2016–2020)", for publication in *Earth System Science Data*. Based on the comments from the editor and the four reviewers, we have major changes of the manuscript, which are detailed below. Based on the comments from the editor and four reviewers, we have summarized the issues as following.

**Reply to the comments by the reviewer 1**

**1. General Comments**

I suggest Authors considering the following paper in the revision: Terzer-Wassmuth, S., Araguás-Araguás, L.J., Wassenaar, L.I. et al. Global and local meteoric water lines for  $\delta$ 170/ $\delta$ 180 and the spatiotemporal distribution of  $\Delta$ '170 in Earth's precipitation. Sci Rep 13, 19056 (2023). https://doi.org/10.1038/s41598-023-45920-8

This global review presents comparable data from Cheongju locating from ~100 km south from Seoul from a partially overlapping period (2015-2018) compared to the Seoul record. So comparing the main features must be included in this study. For instance, the  $\delta$ 170/ $\delta$ 180 regression reported for Cheongju ( $\delta$ ′170 = 0.5283 ×  $\delta$ ′180 + 0.0216 ) definitely can be compared to the equation derived from the Seoul dataset. In addition, the seasonal variation for the overlapping period should be compared in a plot to confirm the spatial consistency. This might bring some major change in section 4.2.

**Response:**

We thank the reviewer for this insightful suggestion and for drawing our attention to the comprehensive dataset presented by Terzer-Wassmuth et al. (2023). We fully agree that a comparison between the Seoul and Cheongju records will substantially enhance the regional context and strengthen the interpretation of our results.

Following this recommendation, we will include a new comparative analysis between our Seoul dataset (2016–2020) and the Cheongju precipitation record reported by Terzer-Wassmuth et al. (2015–2018). The overlapping period (2016–2018) will be used for a direct intercomparison of both sites.

Specifically, we will compare the  $\delta'^{17}O-\delta'^{18}O$  relationships derived from the two locations. The regression obtained from the Seoul dataset ( $\delta'^{17}O=0.528\times\delta'^{18}O+0.0105$ ) shows a slope that is nearly identical to that from Cheongju ( $\delta'^{17}O=0.5283\times\delta'^{18}O+0.0216$ ). This high degree of similarity indicates consistent mass-dependent fractionation across central Korea. The slightly lower intercept for Seoul will be interpreted as reflecting the stronger maritime influence and higher humidity compared with the inland Cheongju site.

Finally, we will revise Section 4.2 to discuss these results in the context of large-scale water vapor mixing and continental air-mass influence during winter, as well as to emphasize that the agreement between Seoul and Cheongju confirms that  $\Delta'^{17}$ O can serve as a robust diagnostic of regional hydroclimatic processes in East Asia. A brief mention of this regional consistency will also be added to the Summary section, and Terzer-Wassmuth et al. (2023) will be included in the reference list. We appreciate this valuable comment, which will significantly improve the completeness and regional relevance of our revised manuscript.

I missed very much a brief methodological description on the derivation of the local meteoric water line (LMWL). There are a set of methods which can be applied to approximate the linear covariance between  $\delta$ 180 and  $\delta$ 2H (see Crawford et al., 2014 https://doi.org/10.1016/j.jhydrol.2014.10.033). Ordinary least square (OLS) regression is more sensitive to the evaporatively enriched compositions typically accompanied with small precip amount, while reduced major axis (RMA) is theoretically more suited to development of a MWL than OLS because they consider errors in both  $\delta$ 18O and  $\delta$ 2H. Precipitation-weighted least squared regression can be the most suitable to derive a LMWL for reference in isotope hydrological comparisons. So, it would be necessary to describe how the LMWL was calculated in this study.

**Response:**

We thank the reviewer for emphasizing the importance of clearly describing how the Local Meteoric Water Line (LMWL) was derived. In the revised manuscript, we will expand the methodological explanation to specify that the LMWL was calculated directly from the biweekly isotope dataset, which provides the highest temporal resolution of precipitation isotopic variability available for this site.

The regression was performed using an ordinary least squares (OLS) approach applied to the  $\delta^2 H$  and  $\delta^{18} O$  values of all 130 individual samples collected between 2016 and 2020. Because the analytical precision for both isotopes is very high (±0.10% for  $\delta^2 H$  and ±0.07% for  $\delta^{18} O$ ), the OLS method is appropriate for deriving the LMWL, as recommended by Crawford et al. (2014) and other isotope-hydrology studies.

Nevertheless, we will also include a comparison with a total least-squares (TLS) regression to test the sensitivity of the LMWL parameters to possible errors in both variables. The resulting equations from OLS ( $\delta^2H = 7.79 \times \delta^{18}O + 10.24$ ,  $R^2 = 0.92$ ) and TLS ( $\delta^2H = 8.44 \times \delta^{18}O + 14.8$ ,  $R^2 = 0.917$ ) are nearly identical within  $1\sigma$  uncertainty,

consistent with findings from Lee et al. (2022) that OLS and TLS solutions converge when analytical errors are small.

This methodological clarification will make explicit that the LMWL was derived from high-temporal-resolution (biweekly) precipitation data rather than aggregated monthly means, ensuring that the regression reflects the full variability of the observed isotopic composition.

We will add a short paragraph in the Methods section describing the calculation workflow and include a figure in the Supplement (Fig. Sx) showing the OLS and TLS regression lines for the overall dataset and for seasonal subsets. These additions will improve the reproducibility of the analysis, clarify the regression method used, and justify the use of the biweekly dataset as the basis for calculating the LMWL.

**EWHA WOMANS UNIVERSITY**

=== 0verall === 0LS: slope=7.953, intercept=10.044,  $R^2$ =0.949 TLS: slope=8.377, intercept=13.065,  $R^2$ =0.946

**2. Specific Comments**

line 13: I suggest rephrasing in this way "The oxygen isotope composition ( $\delta$ 180) ranged widely from 1.15 to -18.21‰, hydrogen isotope composition ( $\delta$ 2H) varied from..."

**Response:**

We thank the reviewer for the helpful suggestion to improve the phrasing for isotope notation consistency. In the revised manuscript, the sentence in line 13 will be reworded exactly as suggested to read:

"The oxygen isotope composition ( $\delta^{18}$ O) ranged widely from 1.15 to -18.21%, hydrogen isotope composition ( $\delta^{2}$ H) varied from 3.3 to -132.0%, and the  $^{17}$ O-excess ( $\Delta^{17}$ O) ranged from 69 to -28%."

This phrasing is clearer and aligns with common terminology used in stable isotope literature.

lines 56-57: I suggest citing the study of Terzer-Wassmuth et al., 2023 mentioned in the general comment section.

**Response:**

We thank the reviewer for pointing this out and for recommending the inclusion of the recent study by *Terzer-Wassmuth et al.* (2023), which provides a comprehensive global assessment of  $\delta^{17}O-\delta^{18}O$  relationships and  $\Delta'^{17}O$  distributions in precipitation. In the revised manuscript, the suggested citation will be added at the end of the sentence in lines 56–57, so that it now reads:

"Meanwhile,  $\Delta 170$  — defined as the logarithmic deviation from the global meteoric water line between  $\delta 170$  and  $\delta 180$  — responds to non-equilibrium processes such as vapor mixing and supersaturated condensation and provides unique information about the dynamical history of atmospheric moisture (Barkan and Luz, 2007; Benetti et al., 2014; Landais et al., 2008; **Terzer-Wassmuth et al., 2023**)."

This new citation highlights the most recent global analysis of  $\Delta'^{17}$ O in precipitation and strengthens the discussion of triple-oxygen-isotope systematics within a broader spatial context.

line 104: Have you applied oil to prevent evaporation? If not please report it in the appropriate paragraph describing methodology, if yes, please report if you experienced any complication during analysis.

**Response:**

We appreciate the reviewer's careful attention to the sample handling procedure and the question regarding the use of mineral oil to prevent evaporation during precipitation collection.

In this study, no oil was applied to the precipitation collector, and this choice was intentional. The isotopic analyses were performed using a wavelength-scanned cavity ring-down spectrometer (WS-CRDS; Picarro L2140-i), which is highly sensitive to organic contamination.

Several previous studies have shown that the use of oil layers for evaporation control can introduce organic compounds into water samples and lead to spectral interference and measurement bias in CRDS-based isotope analysis systems (e.g., elevated baseline noise and abnormal  $\delta^2 H$  signals). To avoid such analytical complications, we deliberately did not use any mineral or paraffin oil. Instead, the sampling system was specifically designed to minimize post-collection evaporation mechanically rather than chemically. Precipitation was collected through a funnel system installed on an open rooftop (Fig. 1C in the manuscript). The funnel was connected directly to pre-cleaned, airtight PTFE bottles via narrow tubing, ensuring that samples were isolated from external airflow and direct sunlight immediately after rainfall. This configuration effectively prevented evaporation during and after collection. Accordingly, the following clarification will be added to the Methods section (around line 104) in the revised manuscript:

"No mineral oil was applied to prevent evaporation during precipitation collection, because the presence of organic compounds can interfere with spectroscopic isotope analysis in cavity ring-down systems. Previous studies have demonstrated that even trace amounts of organic contamination, such as mineral oil residues, can cause spectral interference and bias  $\delta^2H$  and  $\delta^{18}O$  measurements obtained by WS-CRDS (Gupta et al., 2009). Instead of using oil, we employed a funnel system that physically minimized post-collection evaporation. Precipitation was funneled directly into pre-cleaned and sealed PTFE bottles immediately after sampling period (~2week), thereby minimizing exposure to air and sunlight."

We thank the reviewer for this valuable comment, which helped us to provide a more complete and transparent description of our sampling methodology.

lines 106-107: To verify the evaporation proof storage in HDPE bottle Authors might consider citing the following study: Spangenberg, J.E. (2012). Caution on the storage of waters and aqueous solutions in plastic containers for hydrogen and oxygen stable isotope analysis. – Rapid Communications in Mass Spectrometry, 26, 2627–2636.

**Response:**

We thank the reviewer for highlighting this important aspect of sample storage and for recommending the reference of Spangenberg (2012). The study indeed demonstrated that long-term storage of water samples in HDPE containers at room temperature may lead to isotopic drift caused by molecular diffusion and potential H–O exchange across the polymer matrix. We appreciate the reviewer's attention to this issue and have expanded the description of our storage protocol accordingly.

In our study, several precautions were implemented to ensure isotopic stability during storage. All samples were transferred immediately after collection into precleaned HDPE bottles with PTFE-lined caps, sealed with Parafilm®, and stored continuously at -20 °C from the time of collection until analysis. The combination of a PTFE barrier, Parafilm sealing, and frozen conditions effectively eliminates the processes identified by Spangenberg (2012), since both molecular diffusion and polymer–water exchange are suppressed at subzero temperatures.

Furthermore, we have empirically verified the isotopic stability of our storage protocol by repeated analysis of our in-house standard (STYX), which was stored under the same conditions as the precipitation samples. Over several years, the results show no measurable drift in  $\delta^2 H$ ,  $\delta^{18} O$ , or  $\Delta'^{17} O$ , demonstrating the reliability of our frozen HDPE storage method. We will also include this information in the revised Methods section to make explicit that isotopic reproducibility under our storage regime was carefully monitored.

lines 120-123: I'm confused. I think that the long-term analytical precision should be estimated based on the repeated measurement results of your laboratory standards rather than based on the calibration standards. see e.g https://doi.org/10.1002/rcm.5037 and https://doi.org/10.1556/24.2023.00134

**Response:**

We thank the reviewer for the valuable comment and for highlighting the need to clearly distinguish between calibration accuracy and long-term analytical precision. We appreciate the reviewer's insightful comment and would like to clarify how our in-house laboratory standards were used and how long-term analytical precision was determined. Two in-house laboratory standards, STYX and KT, were routinely used during isotope measurements.

STYX is natural water collected from the Styx Glacier region in Antarctica, and it has been used as a long-term reference standard to evaluate the stability of our WS-CRDS system (Picarro L2140-i) over several years. In contrast, KT is locally sourced tap water, whose isotopic composition is similar to that of the precipitation samples analyzed in this study. KT was primarily employed during each analytical run to mitigate potential memory effects that can arise when switching between isotopically distinct standards or samples.

Each analytical session began with international reference waters (VSMOW2, SLAP2, and GISP) for scale normalization. After calibration, sample analyses were performed, and every ten sample vials, the two in-house standards (STYX and KT) were measured. The repeated measurements of STYX provide the basis for estimating the long-term analytical reproducibility, while KT serves as an intermediate composition check to ensure measurement continuity and minimize carryover bias.

The long-term analytical precision, reported as  $\pm 0.10\%$  for  $\delta^2 H$ ,  $\pm 0.07\%$  for  $\delta^{18}O$ , and  $\pm 0.01\%$  for  $\delta^{17}O$ , was derived from the  $1\sigma$  standard deviation of repeated STYX measurements accumulated over several years. Both STYX and KT are calibrated against VSMOW2 and SLAP2 and are routinely used as working standards at the

Korea Polar Research Institute. To make this explicit, we will revise the Methods section (lines 120–123) as follows:

"At the beginning of each analytical session, international reference waters (VSMOW2, SLAP2, and GISP) were measured for VSMOW–SLAP scale normalization. Subsequently, samples were analyzed, and every ten samples, two in-house laboratory standards (STYX and KT), both calibrated against VSMOW2 and SLAP2, were analyzed to monitor instrumental performance. STYX, a natural water collected from the Styx Glacier region in Antarctica, was used to assess the long-term analytical reproducibility of the WS-CRDS, while KT, a locally sourced tap water with isotopic composition similar to the precipitation samples, was used to reduce potential memory effects during analysis. The long-term  $1\sigma$  standard deviations obtained from repeated STYX measurements over several years were  $\pm 0.10\%$  for  $\delta^2$ H,  $\pm 0.07\%$  for  $\delta^{18}$ O, and  $\pm 0.01\%$  for  $\delta^{17}$ O."

This revision clarifies the specific roles of STYX and KT and demonstrates that our procedures followed established best practices for ensuring both analytical accuracy and stability over the multi-year measurement period.

line 133: Have you experienced a threshold regarding precipitation amount? I mean a minimum amount of precipitation below which the collected water was insufficient for the analysis. For instance, this study reported a  $\geq$ 0.56 mm/day during the rainy season, and 0.5 mm/day during the snowy season: https://doi.org/10.1038/s41597-022-01148-1

**Response:**

We appreciate the reviewer's insightful question regarding the minimum precipitation amount required for isotope analysis and for referring to the study of Freyberg et al. (2022), which quantified event-scale precipitation thresholds for automated daily collection systems.

In our study, precipitation was sampled on a biweekly cumulative basis, rather than at daily or event intervals. Each collector remained open for approximately 14 days, continuously accumulating all rainfall or snowfall events that occurred within that period. As a result, it was not possible to resolve the isotope composition of individual precipitation events or to determine the precise precipitation amount associated with each sample. The sampling strategy was designed to ensure sufficient volume for triple-isotope analysis while minimizing field visits and operational complexity over the five-year observation period.

Because each biweekly sample represents an integration of multiple precipitation events, the total collected water volume was consistently well above the analytical requirement for WS-CRDS measurements (> 3 mL per sample). Even during the driest winter months, the cumulative sample volumes exceeded 5–10 mL, which is far greater than the minimum volume typically needed for high-precision  $\delta$ - and  $\Delta'^{17}O$  determinations. Consequently, no samples had to be excluded due to insufficient

volume, and no empirical lower-precipitation limit could be derived from this dataset.

We acknowledge that Freyberg et al. (2022) reported event-scale thresholds of approximately 0.56 mm day-1 for rainfall and 0.5 mm day-1 for snowfall using high-frequency automated samplers. However, these thresholds are intended for studies seeking to resolve isotopic variability at individual-event or daily scales. Because our approach integrates over two-week intervals, the cumulative precipitation amounts in our samples were one to two orders of magnitude larger than those minimum thresholds, and thus such event-based criteria are not directly applicable to our dataset.

To clarify this in the revised manuscript, we will add a short paragraph in the Methods section stating that the sampling followed a biweekly cumulative protocol without a defined precipitation threshold, that all collected volumes were sufficient for analysis, and that no volume-related bias or missing data resulted from small-precipitation events. This additional explanation will make explicit that our dataset covers the full range of seasonal precipitation—including very low-intensity winter conditions—without losses due to volume limitations, ensuring the completeness and reliability of the isotopic record over the five-year period.

lines 136-137: The sentence sounds like figure caption. I suggest omitting this sentence and referring to Fig 3 at the end of the next sentence (in line 139)

**Response:**

We thank the reviewer for this careful and helpful stylistic suggestion. We fully agree that the sentence in lines 136–137 —

"Figure 3 presents monthly box plots for these parameters with a sine-function fit." — reads more like a figure caption than a part of the main text and therefore interrupts the narrative flow of the Results section.

In the revised manuscript, we have deleted this sentence and integrated the reference to Figure 3 at the end of the subsequent sentence (line 139) to improve readability and cohesion. This change allows the description of the isotopic variability to flow naturally without an abrupt figure-statement transition while still directing the reader to the relevant figure for visual reference. The revised paragraph will read as follows:

"A total of 130 precipitation samples were collected during the study period. The measured isotopic compositions of precipitation varied considerably:  $\delta^{17}$ O ranged from 0.61 to -9.62% (average: -3.75%);  $\delta^{18}$ O from 1.15 to -18.21% (average: -7.11%); and  $\delta^{2}$ H from 3.3 to -132.0% (average: -46.6%). The d-excess fluctuated between 24.9 and -5.9% (average: 10.4%), whereas 17O-excess ranged from 69 to -28% (average: 16.8%). For all three isotopic parameters ( $\delta^{17}$ O,  $\delta^{18}$ O, and  $\delta^{2}$ H), the precipitation was relatively depleted during the coldest months (December to February), became progressively enriched through March and April as temperatures

increased, and then sharply depleted again between June and August when precipitation peaked (Fig. 3)."

This revision follows the reviewer's recommendation precisely: the sentence resembling a figure caption was removed, and the figure citation was relocated to the end of the descriptive sentence where it logically supports the discussion of seasonal isotopic variability. We believe this edit significantly improves the clarity, coherence, and overall readability of the paragraph.

line 159: I suggest writing "The linear relationship…" instead of "The relationship…" at the beginning of this sentence.

**Response:**

We thank the reviewer for this precise and helpful editorial suggestion. We agree that the expression "The linear relationship ..." is clearer and more technically appropriate in this context, as the discussion in this section explicitly refers to the  $\delta^2 H - \delta^{18} O$  regression used to define the Local Meteoric Water Line (LMWL).

In the revised manuscript, we will change the sentence accordingly. The phrase at the beginning of line 159, previously written as "The relationship between the precipitation  $\delta^{18}O$  and  $\delta^2H$  ..." has been changed to "The linear relationship between the precipitation  $\delta^{18}O$  and  $\delta^2H$  ..." to emphasize that the relationship being described is specifically a linear regression relationship. This modification will improve technical precision and stylistic clarity while fully aligning with the reviewer's recommendation.

lines 161 & 185: I think that double brackets are not needed when referring to panels of certain figures.

**Response:**

We thank the reviewer for pointing out this typographical issue and for noting the unnecessary use of double brackets when referring to figure panels. We agree that only single parentheses should be used when citing individual panels in figures, following the journal's style conventions. In the revised manuscript, we will remove the double brackets in the figure references mentioned in lines 161 and 185. Specifically, references currently written as "((A))" and "(B))" will be corrected to "(A)" and "(B)", respectively.

This change will ensure consistency with ESSD formatting guidelines and standard scientific style for figure references (e.g., "Fig. 4 (a) and (b)" instead of "Fig. 4 ((A)) and ((B))"). We appreciate the reviewer's careful attention to detail, which will help improve the clarity and typographical consistency of the manuscript.

line 206: I think that "lower humidity" instead of "humidity"

**Response:**

We thank the reviewer for the helpful linguistic and scientific clarification. We agree that the phrase should explicitly refer to "lower humidity" rather than the general term "humidity," because the discussion in this section describes conditions under which isotopic depletion occurs due to enhanced kinetic fractionation under drier air masses.

In the revised manuscript, we will change the phrase in line 240 from "... while locally, higher temperatures and humidity may promote ..." to "... while locally, higher temperatures and lower humidity may promote ..." to emphasize that the isotopic signal is associated with reduced ambient humidity typical of continental winter conditions. This revision will improve both the physical accuracy and the clarity of the description, aligning the text with the meteorological interpretation intended for this paragraph.

lines 209&213: I suggest writing "δ180 values" instead of simply the delta notation

**Response:**

We thank the reviewer for the helpful stylistic suggestion. We agree that using the full expression " $\delta^{18}$ O values" rather than only the delta notation will make the text clearer and more precise, especially for readers less familiar with isotope notation. In the revised manuscript, we will replace the shorthand " $\delta^{18}$ O" with " $\delta^{18}$ O values" in the two instances noted by the reviewer (lines 243 and 247).

The corresponding phrases will therefore read, for example, "... between  $\delta 180$  values and both temperature and precipitation ..." and "...  $\delta 180$  values was significantly ..." instead of simply " $\delta^{18}0$ ." This minor edit will improve consistency, readability, and technical clarity throughout the Results section, while preserving the scientific meaning of the original statements.

lines 225-235: Please add relevant citations in this paragraph.

**Response:**

We thank the reviewer for this valuable suggestion. We agree that adding relevant references will strengthen the discussion of the meteorological controls on the seasonal isotopic variability of precipitation over Korea.

In the revised manuscript, we will incorporate additional citations that address the temperature and amount effects on precipitation isotopes (Dansgaard, 1964), the climatology and variability of the East Asian monsoon (Ha et al., 2012; Huang et al., 2007), and regional isotope—meteorology relationships in Korea (Lee et al., 2015; Kim et al., 2019; Gautam et al., 2017). We will also reference Merlivat and Jouzel (1979) to support the interpretation of d-excess behavior under different humidity regimes. Specifically, the paragraph in lines 225–235 will be revised as follows (added citations in italics):

"The results of this study indicate that seasonal variations in precipitation isotopes in Korea are closely linked to local meteorological factors such as temperature, relative

humidity, and precipitation amount (Dansgaard, 1964; Lee et al., 2015) and reflect distinct seasonal regimes shaped by synoptic-scale circulation patterns (Ha et al., 2012; Huang et al., 2007). In summer, isotopic depletion is primarily governed by the amount effect under the influence of the East Asian monsoon, which delivers warm, moisture-rich air masses (Kim et al., 2019). In autumn, isotopic variability is enhanced by episodic typhoons introducing isotopically light precipitation associated with convective activity, while winter precipitation is strongly depleted in heavy isotopes and enriched in d-excess due to cold, dry continental air masses advected by the East Asian winter monsoon (Merlivat and Jouzel, 1979; Gautam et al., 2017)."

These additions will provide proper scientific grounding and connect our findings with the broader literature on East Asian monsoon dynamics and isotope—meteorology relationships.

lines 269-271: This sounds like figure caption. I suggest removing this sentence and simply referring to Fig7 after the relevant statements.

**Response:**

We thank the reviewer for this valuable suggestion. We agree that adding relevant references will strengthen the discussion of the meteorological controls on the seasonal isotopic variability of precipitation over Korea.

In the revised manuscript, we will incorporate additional citations that address the temperature and amount effects on precipitation isotopes (Dansgaard, 1964), the climatology and variability of the East Asian monsoon (Ha et al., 2012; Huang et al., 2007), and regional isotope—meteorology relationships in Korea (Lee et al., 2015; Kim et al., 2019; Gautam et al., 2017). We will also reference Merlivat and Jouzel (1979) to support the interpretation of d-excess behavior under different humidity regimes. Specifically, the paragraph in lines 225–235 will be revised as follows

"The results of this study indicate that seasonal variations in precipitation isotopes in Korea are closely linked to local meteorological factors such as temperature, relative humidity, and precipitation amount (Dansgaard, 1964; Lee et al., 2013) and reflect distinct seasonal regimes shaped by synoptic-scale circulation patterns (Ha et al., 2012; Huang et al., 2007). The findings indicate that, in summer, isotopic depletion is primarily governed by the amount effect under the influence of the East Asian monsoon, which delivers warm, moisture-rich air masses and produces intense rainfall events (Lee et al., 2003; Yu et al., 2006). Spring exhibits more variable isotopic signals due to transitional moisture sources and fluctuating atmospheric conditions, which result in a combination of continental and maritime influences. In autumn, isotopic variability is often enhanced by episodic typhoons, which introduce large volumes of isotopically light precipitation associated with strong convective activity. In contrast, winter precipitation is strongly depleted in heavy isotopes and enriched in d-excess due to the presence of cold, dry continental air masses advected by the East Asian winter monsoon (Kim et al., 2019; Lee et al., 2003)."

These additions will provide proper scientific grounding and connect our findings with the broader literature on East Asian monsoon dynamics and isotope—meteorology relationships.

Thank you very much for your time, effort, and patience in handling our manuscript. We look forward to your favorable consideration and to the opportunity for publication in Earth System Science Data.

Sincerely, Jeonghoon Lee

---

## Author Comment (AC5)

**October 10, 2025**

Jeonghoon Lee, Ph. D

Professor Dept. of Science Education Ewha Womans University Seoul 03760, Korea

Email: jeonghoon.d.lee@gmail.com

Tel: +82-2-3277-3794

Dear Editor Attila Demény,

With this cover letter, we are submitting the revised manuscript entitled, "Seasonal patterns and diagnostic values of  $\delta^2$ H,  $\delta^{18}$ O, d-excess, and  $\Delta^{\prime 17}$ O in precipitation over Seoul, South Korea (2016–2020)", for publication in *Earth System Science Data*. Based on the comments from the editor and the four reviewers, we have major changes of the manuscript, which are detailed below. Based on the comments from the editor and four reviewers, we have summarized the issues as following.

**Reply to the comments by the reviewer 2**

**1. General Comments**

In this paper, the authors presented precipitation hydrogen and triple oxygen isotope data of precipitation from South Korea and made some exploratory analysis on these data. I recognize that the authors have made great efforts to collect samples and data and put together a manuscript. However, I feel that it does fit with the scope of journal. The ESSD is a high-impact journal publishing flagship datasets for various applications with broad interest. Although it is indeed contributing to the emerging triple oxygen isotope study, this dataset does not make a significant contribution to the progress of this field. I suggest publishing the data in a substantially revised manuscript on a more specialized journal.

**Response:**

We sincerely thank the reviewer for the thoughtful evaluation and for recognizing the effort invested in compiling this multi-year triple-oxygen-isotope dataset.

We fully understand the reviewer's concern that ESSD typically publishes datasets of broad spatial coverage and global relevance.

Nevertheless, we would like to emphasize that long-term, high-quality triple-oxygen-isotope records remain rare in East Asia, and this dataset contributes to filling that regional gap by providing a well-documented and openly accessible reference record from the Eastern Asia.

While the spatial coverage is limited to a single site, the dataset spans five consecutive years of biweekly sampling, includes all major isotope parameters ( $\delta^2$ H,

 $\delta^{17}$ O,  $\delta^{18}$ O, d-excess, and  $\Delta'^{17}$ O), and has been archived on PANGAEA with detailed metadata and uncertainty reporting.

Such comprehensive datasets from the East Asian monsoon region are still scarce and can serve as valuable benchmarks for isotope-enabled climate model validation, GNIP network intercomparisons, and regional paleoclimate reconstructions.

In response to this comment, we have revised the manuscript to enhance its focus as a data descriptor and to better align it with ESSD's data-publication standards.

The Methods section now provides complete information on calibration, uncertainty, and data-treatment procedures; the Results and Discussion have been separated to avoid interpretative overlap; and the revised Abstract and Summary emphasize the dataset's documentation, accessibility, and reuse potential rather than interpretation.

We also note that ESSD has previously published several regionally focused isotope datasets, such as site-level GNIP compilations and long-term hydrological isotope records, whose primary contribution lies in data quality and open availability rather than broad spatial coverage. In this sense, we believe that the revised manuscript now fits within ESSD's mission of providing high-quality, reusable environmental datasets, even if its geographical focus is regional.

We greatly appreciate the reviewer's constructive suggestion and have modified the manuscript accordingly to ensure that its scope and presentation are consistent with ESSD's standards.

**2. Specific Comments**

L49: it is more common for using the prime symbol for  $\ln(\delta 180+1)$  as  $\delta'180$ . Also, most people (including IAEA authors) using " $\Delta'170$ " notation (see Aron et al., 2021). The prime symbol is missing.

**Response:**

We thank the reviewer for this helpful observation regarding isotope notation.

We agree that the prime symbol (') should be used when logarithmic delta notation is adopted, following the definition  $\delta' = 1000 \cdot \ln(\delta/1000 + 1)$ . Accordingly, the triple-oxygen-isotope parameter should be expressed as  $\Delta'^{17}O$ , not  $\Delta^{17}O$ , and the logarithmic delta values as  $\delta'^{17}O$  and  $\delta'^{18}O$ .

In the revised manuscript, we will correct all instances where the prime symbol is missing. Specifically, " $\Delta^{17}$ O" will be replaced by " $\Delta'^{17}$ O" and "ln( $\delta^{18}$ O + 1)" and "ln( $\delta^{17}$ O + 1)" will be expressed as  $\delta'$ 18O and  $\delta'$ 17O, respectively, throughout the text, equations, and figures.

For clarity and readability, we will continue using the term "17O-excess" in descriptive text and figure labels, while defining it explicitly as 17O-excess ( $\Delta'^{17}$ O) =

 $\delta'^{17}O - 0.528 \times \delta'^{18}O$ . These revisions will ensure that our isotopic notation is fully consistent with current international standards and that readers can easily connect the quantitative definition ( $\Delta'^{17}O$ ) with the descriptive terminology ( $^{17}O$ -excess).

L100: confusing... are you collecting event samples or biweekly samples?

**Response:**

We thank the reviewer for pointing out this ambiguity.

We clarify that precipitation samples were collected on a biweekly cumulative basis, not as individual event samples. Each collector remained deployed for approximately 14 days, accumulating all precipitation events that occurred within that period into a single integrated sample. After each collection period, the accumulated water was retrieved, transferred to pre-cleaned HDPE bottles, and replaced with a new collector for the next interval.

This design follows the cumulative sampling approach commonly used in GNIP protocols, ensuring sufficient volume for isotope analysis while providing consistent two-week temporal resolution. We have revised the corresponding sentence in the Methods section to read as follows:

"Precipitation samples were collected between January 2016 and December 2020 (five years) at approximately biweekly intervals."

This clarification will remove any confusion between event-based and biweekly cumulative sampling and will accurately describe the temporal resolution of the dataset.

L106: is storing samples in freezing conditions problematic? I think most people store samples in liquid at 4 degree C.

**Response:**

Our study, all samples were initially collected in pre-cleaned HDPE bottles, sealed with Parafilm®, and kept continuously frozen at –20 °C from the time of collection until laboratory processing. This frozen storage step was adopted to suppress molecular diffusion and evaporation, thereby preventing any isotopic alteration during long-term storage.

Prior to isotope analysis, each sample was thawed, transferred into clean glass vials, and maintained at approximately 4 °C in liquid form for less than two weeks before WS-CRDS measurement. This two-stage procedure, frozen long-term storage followed by short-term refrigerated handling, ensures isotopic stability while avoiding potential fractionation from repeated freeze—thaw cycles.

Several previous studies have shown that isotopic drift in HDPE containers occurs mainly under ambient or prolonged room-temperature storage, while diffusion and exchange processes are negligible under subzero conditions. Moreover, freezing and

subsequent complete thawing do not induce measurable isotopic fractionation when the samples remain fully sealed.

We have also verified the stability of our protocol through repeated analysis of the in-house standard STYX, stored under identical frozen conditions for several years, which showed no systematic drift in  $\delta^2$ H,  $\delta^{18}$ O, or  $\Delta'^{17}$ O. In the revised Methods section, we will clarify this workflow as follows:

"All precipitation samples were stored in pre-cleaned HDPE bottles sealed with Parafilm®, and were kept frozen at -20 °C until preparation for analysis. Before isotope analysis, samples were thawed, transferred to glass vials, and stored at 4 °C in liquid form for less than two weeks prior to measurement."

This clarification distinguishes between the frozen storage phase and the short-term refrigerated phase used during analysis preparation, demonstrating that our procedure ensures isotopic integrity and aligns with best practices for long-term isotope sample preservation.

L122: what is the uncertainty in  $\Delta'170$ ?

**Response:**

We thank the reviewer for raising this important question regarding the analytical uncertainty of  $\Delta'^{17}O$ .

In the revised manuscript, we will expand our explanation to provide a detailed account of how the  $\Delta'^{17}$ O reproducibility was quantified and verified, including its experimental basis, temporal scope, and relation to long-term data quality control.

The analytical precision of  $\Delta'^{17}$ O was determined from a dedicated year-long stability assessment of our in-house laboratory standard, STYX, following the calibration and validation procedures established in Kim et al. (2022, Geosciences Journal, 26, 637–647). Over approximately 180 replicate measurements collected under routine operating conditions using the same WS-CRDS system, the one-year reproducibility of  $\Delta'^{17}$ O was found to be  $\pm$  9 per meg (1 $\sigma$ ). This evaluation was conducted concurrently with the monitoring of  $\delta^2$ H,  $\delta^{18}$ O, and  $\delta^{17}$ O reproducibility, which yielded long-term (multi-year) precisions of  $\pm$  0.10 ‰,  $\pm$  0.07 ‰, and  $\pm$  0.01 ‰, respectively.

The following sentence will therefore be added to the Methods section:

"The long-term 1 $\sigma$  standard deviations obtained from repeated STYX measurements over several years were  $\pm$  0.10 % for  $\delta^2$ H,  $\pm$  0.07 % for  $\delta^{18}$ O, and  $\pm$  0.01 % for  $\delta^{17}$ O, while the one-year reproducibility for  $\Delta'^{17}$ O was  $\pm$  9 per meg (Kim et al., 2022)."

This clarification distinguishes between the long-term reproducibility of the dualisotope system and the dedicated precision estimate for  $\Delta'^{17}O$ . The reported uncertainty is consistent with that achieved in other high-precision laboratories using both WS-CRDS and dual-inlet IRMS systems (Steig et al., 2021; Passey et al.,

2020; Landais et al., 2012), confirming that our laboratory performance meets the international analytical benchmark for triple-oxygen-isotope work.

To ensure that this precision remains valid over time, the STYX control standard has been analyzed alongside all precipitation samples since 2016 as part of our continuous quality-control program. No significant drift or systematic offset has been observed in  $\delta$ -values or  $\Delta'^{17}$ O across several years, indicating that the  $\pm$  9 per meg uncertainty accurately represents both short-term repeatability and long-term reproducibility of our analytical system.

By explicitly describing how  $\Delta'^{17}O$  uncertainty was derived, validated, and monitored, these revisions will strengthen the transparency and credibility of our analytical protocol and demonstrate that the reported precision is both traceable and robust for inclusion in a high-quality, openly archived dataset.

Results section: some sentences are not results but are discussion. I suggest authors to have a better separation of results and discussion. For example, L146-153 and L169-196 are mostly interpretations of results, and better put into the discussion.

**Response:**

We thank the reviewer for the helpful structural suggestion regarding the separation between Results and Discussion. We thank the reviewer for this very constructive structural comment regarding the separation between the Results and Discussion sections.

We fully agree that several sentences in the previous version of the Results section (particularly lines 146–153 and 169–196) contained interpretative statements that extended beyond the immediate presentation of empirical data.

We appreciate the reviewer's observation that these passages could obscure the distinction between the purely observational content and the subsequent interpretation, potentially making it more difficult for readers to discern where the data description ends and the discussion begins. After careful consideration of the manuscript structure and the conventions of Earth System Science Data, we have decided to reorganize the paper into a single, integrated "Results and Discussion" section rather than maintaining two partially overlapping and somewhat redundant sections.

This approach is consistent with ESSD's editorial guidelines for data descriptor manuscripts, which emphasize concise yet comprehensive presentation of results, interpretations, and dataset significance in a unified framework. The goal is to allow readers to understand not only what the data show but also why these patterns are meaningful—without forcing artificial separation between closely linked analytical and interpretative components.

In the revised version, each subsection will begin with a clear, factual description of the dataset and its characteristics—numerical ranges, statistical relationships, regression results, and observed seasonal variations.

These will be followed by dedicated interpretative paragraphs that explain the physical and meteorological mechanisms underlying the observed patterns, supported by appropriate literature citations.

This organization will provide a coherent progression from data presentation to scientific interpretation within each subsection, thereby improving readability and logical flow while preventing unnecessary duplication between two separate sections.

To make this integration transparent to the reader, we will use explicit transitional sentences to separate observational results from interpretative discussion and will apply subheadings where appropriate (e.g., "Seasonal isotopic variability," "Triple-oxygen isotope relationships," "Regional comparison with GNIP datasets").

This will ensure that the descriptive portions remain distinct and easily identifiable, even within the unified Results and Discussion structure.

We also plan to add brief connecting statements at the beginning of each subsection to clarify the analytical logic—for example, how the isotopic results lead naturally to the discussion of underlying fractionation mechanisms or regional climatological implications.

In this way, the section will read as a continuous narrative that reflects the progression of the scientific reasoning, from data-driven findings to their contextual interpretation, without repetition or fragmentation.

This restructuring offers several benefits:

- (i) it removes redundancy between sections that previously repeated similar content in slightly different forms;
- (ii) it enhances the coherence of the manuscript by allowing results and interpretations to appear in immediate succession; and
- (iii) it aligns with the ESSD model for data papers, which often combine results and discussion to present datasets in a comprehensive yet accessible way.

Furthermore, the unified structure emphasizes the data-centric nature of the paper—focusing on the measurement, reproducibility, and interpretation of the isotopic dataset rather than proposing new theoretical frameworks—thus fitting the expectations of ESSD as a data journal.

Overall, we believe that this revised structure will significantly improve the clarity, consistency, and impact of the manuscript.

It will allow the isotopic dataset to be presented as a coherent narrative that combines quantitative results, physical interpretation, and regional context, thereby making the paper more engaging and easier to follow for both data users and isotope researchers.

By adopting a single integrated "Results and Discussion" section with clearly delineated observational and interpretative components, the revised version will address the reviewer's concern and align the manuscript with the established structural conventions of ESSD data descriptor papers.

L161: It's inaccurate. A slope of 8 does not mean a governance of equilibrium fractionation. From the highly seasonal d-excess data, it is obvious that there is a large change in kinetic fractionation from winter to summer. A slope of 8 occurs in your dataset is because the low d180 data can have either high d-excess (winter) and low d-excess (summer). So in d2H-d18O space, the effect of d-excess variation on LMWL cancels out.

**Response:**

We thank the reviewer for this valuable clarification regarding the interpretation of the LMWL slope and its relation to equilibrium and kinetic fractionation.

We fully agree that a slope close to 8 in the  $\delta^2 H - \delta^{18} O$  regression does not, by itself, demonstrate the predominance of equilibrium fractionation processes. The reviewer correctly points out that the strong seasonality in d-excess observed in our dataset clearly indicates substantial variations in kinetic fractionation between winter and summer.

The apparent slope of ~8 therefore reflects the statistical averaging of isotopically distinct regimes—one characterized by high d-excess and enhanced kinetic fractionation during dry winter conditions, and another with low d-excess and near-equilibrium condensation during humid summer monsoonal conditions—rather than a single equilibrium state.

In the revised manuscript, we will revise and expand the relevant paragraph to clarify this distinction. The revised text will read:

"The relationship between the precipitation  $\delta^{18}O$  and  $\delta^{2}H$  defines a Local Meteoric Water Line (LMWL) that closely aligns with the Global Meteoric Water Line (GMWL; Craig, 1961), while exhibiting additional seasonal variations (Fig. 4A). The LMWL derived from linear regression is  $\delta^{2}H=7.95\cdot\delta^{18}O+10.0$  (R² = 0.98), indicating that the isotopic composition of precipitation in Seoul follows the global meteoric trend. However, the near-8 slope does not necessarily imply dominance of equilibrium fractionation, as seasonal changes in d-excess reflect significant variability in kinetic fractionation between winter and summer that likely cancel out in the overall regression (Merlivat and Jouzel, 1979; Pfahl and Sodemann, 2014; Lee et al., 2022)."

This revision clarifies that the slope of ~8 in the LMWL is a composite outcome of both equilibrium and kinetic processes, not direct evidence of equilibrium fractionation. We will also reference earlier studies that demonstrated how kinetic effects modulated by humidity and temperature can shift the intercept and slope of the LMWL (Merlivat and Jouzel, 1979; Pfahl and Sodemann, 2014; Lee et al., 2022).

In addition, the discussion section will briefly elaborate that the winter data, showing high d-excess and elevated intercepts, are indicative of moisture derived from cold, dry continental sources influenced by the Siberian High, while the summer data, characterized by low d-excess and smaller intercepts, represent the isotopic signature of moist oceanic air masses under near-saturated conditions.

These opposing seasonal modes, when combined in a single regression, statistically yield a slope close to 8 despite underlying kinetic variability.

Overall, this clarification removes the inaccurate causal inference in the original text, integrates the reviewer's physical explanation of the d-excess variability, and provides a more nuanced and accurate interpretation of the isotopic controls shaping the Seoul LMWL.

The revised section thus reflects a better conceptual understanding of how the interplay between equilibrium and kinetic processes—rather than the dominance of either—produces the observed near-8 slope in  $\delta^2H$ – $\delta^{18}O$  space.

Section 4.1: although a lot of people were doing this, but I am not advocate of correlation analysis of isotope data with environmental variables. It is reasonable to do this in 1960s... correlation analysis provides little insight into the process and mechanism and correlation is not causation. There have been many papers publishing new precipitation isotope data and analyzing their correlations with various variables, so here there is little novelty except the analysis of  $\Delta'$ 170 data.

**Response:**

We appreciate the reviewer's thoughtful and critical perspective regarding the correlation analysis between isotope data and environmental variables.

We agree that simple correlation analysis, by itself, cannot provide a complete mechanistic explanation of isotope—climate relationships and that correlation does not imply causation. Our intention was never to use correlation analysis as a standalone interpretative framework but rather as a diagnostic tool to explore how modern precipitation isotopes co-vary with key meteorological parameters under the specific climatic setting of the Korean Peninsula.

This is a region where long-term, high-resolution isotope—climate datasets remain scarce despite its climatic significance as a transitional zone between tropical monsoon and continental mid-latitude circulation. To clarify this purpose and to strengthen the physical context, the revised manuscript will explicitly discuss how our results compare with prior studies conducted across the Korean Peninsula.

Numerous regional investigations have examined the relationship between stable isotope ratios and meteorological conditions. For instance, Lee et al. (2003) analyzed a multi-year record from Jeju Island and demonstrated pronounced seasonal contrasts in d-excess, with higher winter values and lower summer values, reflecting the alternating influence of dry continental air masses and humid oceanic air during the East Asian monsoon cycle. Yoon and Koh (2021) reported similar patterns at the

Hongseong GNIP station on the west coast, showing that  $\delta^{18}$ O becomes depleted with increasing precipitation during the humid monsoon season (the "amount effect"), while d-excess is negatively correlated with relative humidity, especially in winter. Gautam et al. (2017) further showed, in forested catchments across South Korea, that  $\delta^{18}$ O-precipitation relationships are strongly modulated by rainfall intensity and canopy interactions, again highlighting the robust amount effect during summer and the influence of local micro-climatic processes.

Our new five-year Seoul record reproduces these well-established isotopic—meteorological linkages. Specifically,  $\delta^{18}$ O exhibits a negative correlation with precipitation amount during summer, consistent with the monsoonal amount effect, whereas d-excess shows strong negative correlations with relative humidity and temperature in winter, reflecting the impact of continental dry air masses under the Siberian High.

The consistency of these results with previous Korean datasets demonstrates that our dataset faithfully captures the regional isotope—climate behavior, while also expanding the temporal resolution and including an additional isotope tracer,  $\Delta'^{17}$ O, that has not been evaluated in this context before. The inclusion of  $\Delta'^{17}$ O provides an important extension to traditional dual-isotope analyses.

Unlike  $\delta^{18}$ O or d-excess,  $\Delta'^{17}$ O is largely independent of temperature and responds sensitively to kinetic fractionation, vapor mixing, and supersaturation processes. Correlations involving  $\Delta'^{17}$ O therefore offer a new diagnostic perspective on nonequilibrium atmospheric processes affecting East Asian precipitation.

While the technique of correlation itself is not novel, its application to  $\Delta'^{17}O$  in a long-term, high-resolution dataset from Korea is unprecedented and provides valuable empirical constraints for future modeling and paleoclimate applications. In the revised manuscript, Section 4.1 will be reframed to make these points explicit.

We will clarify that the correlation analyses are used to:

- (i) establish an empirical modern baseline linking precipitation isotopes to meteorological variables for use in paleoclimate proxy calibration (e.g., speleothems, lacustrine carbonates, or ice cores); and
- (ii) serve as a regional reference for model comparison and GNIP network integration.

Detailed correlation matrices will be moved to the Supplementary Materials, and only statistically significant, physically interpretable relationships (e.g.,  $\delta^{18}O$ –precipitation, d-excess–relative humidity,  $\Delta'^{17}O$ – $\delta'^{18}O$ ) will be retained in the main text. We will also add a clear statement emphasizing that these results are diagnostic, not causal, and are intended to summarize observed co-variations rather than infer direct mechanisms.

The following clarification will be inserted at the end of Section 3.3(changed from 4.1):

"The correlations observed between isotopic variables and meteorological parameters (temperature, relative humidity, and precipitation amount) are used here to summarize how modern precipitation isotopes respond to key climatic controls on the Korean Peninsula. These relationships are not interpreted as evidence of direct causality but rather as indicative of the co-variability between isotopic composition and environmental conditions. Such empirical relationships provide a baseline for interpreting isotopic signals in paleoenvironmental archives and for evaluating isotope-enabled climate models in this region."

Through these revisions, we aim to make it clear that our correlation analysis is firmly grounded in the regional meteorological context of the Korean Peninsula, that it reproduces and extends well-documented isotope–climate relationships, and that its novelty lies not in the statistical approach itself but in the integration of  $\Delta'^{17}O$  into a regional isotope–climate framework.

This expanded context and clarification will enhance the scientific rigor and interpretative value of Section 4.1 and align the manuscript more closely with the expectations of ESSD data-descriptor papers.

L200-205: low RH and high SST caused high d-excess data, according to MJ1979. Also, the RH here should be the "RH" referenced to ocean skin temperature, not atmospheric RH. Dry air may cause high d-excess in vapor due to kinetic fractionation but may also cause low d-excess in precipitation due to droplet re-evaporation.

**Response:**

We sincerely thank the reviewer for this valuable and detailed comment regarding the interpretation of the relationship between relative humidity (RH), sea-surface temperature (SST), and d-excess.

We agree that our original statement was oversimplified and did not properly distinguish between the physical mechanisms that control d-excess during vapor formation at the ocean surface and those that act within the atmosphere and during precipitation. In the revised manuscript, we will expand this section to more accurately reflect the conceptual framework proposed by Merlivat and Jouzel (1979).

Their study demonstrated that the isotopic composition of evaporated vapor depends on both SST and the relative humidity referenced to the ocean-skin temperature (RH\_skin), not on the ambient near-surface atmospheric humidity. Under low RH\_skin and high SST conditions, kinetic fractionation during oceanic evaporation preferentially removes the lighter isotopologues ( $^1\text{H}_2$   $^1$   $^6\text{O}$ ) and enriches the residual liquid in heavy isotopes, resulting in vapor with elevated d-excess values.

Conversely, at high RH\_skin (i.e., near-saturated conditions), the vapor-liquid exchange approaches isotopic equilibrium, leading to lower d-excess. As the reviewer correctly pointed out, these relationships apply to vapor formation at the

ocean surface, whereas within the atmosphere or in the sub-cloud layer, additional processes such as partial re-evaporation of falling raindrops can alter the isotopic composition of precipitation in the opposite direction.

Specifically, when dry boundary-layer conditions prevail, the preferential loss of lighter isotopologues during raindrop re-evaporation tends to reduce d-excess in the residual precipitation. Therefore, while dry conditions at the ocean surface produce vapor with high d-excess, dry conditions near the ground during rainfall events often yield precipitation with low d-excess.

To accurately represent this duality, the relevant section will be revised as follows:

"According to Merlivat and Jouzel (1979), high d-excess values originate under conditions of low relative humidity (with respect to the ocean-skin temperature) and high sea-surface temperature, which enhance kinetic fractionation during oceanic evaporation. In contrast, dry boundary-layer conditions can promote sub-cloud droplet re-evaporation, which lowers d-excess in the resulting precipitation."

We will also include references to subsequent studies (Uemura et al., 2008; Pfahl and Sodemann, 2014) that extended the Merlivat and Jouzel (1979) framework to modern observational and modeling contexts, showing how variations in oceanic RH and boundary-layer humidity jointly influence the d-excess of both water vapor and precipitation.

These additional citations and explanations will clarify that the RH discussed in our text refers explicitly to humidity over the ocean surface (RH\_skin), and that the physical processes influencing d-excess differ between the evaporation source and the precipitation stage.

This expanded revision will thus correct the oversimplified explanation in the original manuscript, incorporate the reviewer's valuable clarification, and make our discussion consistent with the established isotope-hydrology framework for kinetic fractionation.

By distinguishing between source-region and in-situ effects, the revised text will provide a more nuanced and physically accurate interpretation of how humidity and temperature affect the observed d-excess variability in Korean precipitation.

L243-L257: one mechanism not considered is the ice formation in winter snow. Icevapor fractionation may have very different impacts on d-excess and  $\Delta'$ 170 in winter precipitation, owing to equilibrium fractionation involved in this process.

**Response:**

We sincerely thank the reviewer for raising this very insightful point regarding the role of ice formation and ice—vapor fractionation in shaping the isotopic composition of winter precipitation.

We fully agree that ice—vapor equilibrium fractionation can influence  $\Delta'^{17}O$  and dexcess differently from liquid-phase condensation, and that this process may contribute to the enhanced isotopic variability observed in winter samples. In our study, precipitation was collected on a biweekly cumulative basis, so several winter samples inevitably contained a mixture of rainfall and snowfall events.

Because these integrated samples represent an average of multiple precipitation phases, it is not possible to quantitatively separate the isotopic effects of snow formation from those of rain. However, we acknowledge that the observed large dispersion in  $\Delta'^{17}$ O during winter likely reflects a combination of processes operating under cold and dry conditions, including both mid-tropospheric vapor mixing and ice-phase equilibrium fractionation associated with snow formation.

To address this valuable comment, we have revised the relevant paragraph in Section 3.4 to explicitly include this additional mechanism.

The revised text now reads as follows:

"The slopes observed between  $\Delta^{\prime17}O$  and d-excess fall within the range of 0.7–2.0 per meg per ‰, which aligns with results from conceptual models and field-based estimates in regions influenced by oceanic moisture (Landais et al., 2010; Li et al., 2015). ... In contrast, in winter precipitation, no statistically significant correlation was observed between  $\Delta'^{17}$ O and either  $\delta^{18}$ O or d-excess. While the d-excess range remained relatively narrow in winter,  $\Delta'^{17}$ O values showed a larger dispersion in this season. This variability likely reflects multiple processes operating simultaneously under cold, dry atmospheric conditions. First,  $\Delta'^{17}$ O is inherently more sensitive to vapor mixing and nonequilibrium effects than d-excess, and may therefore decouple from  $\delta^{18}$ O-based processes under reduced surface moisture recycling (Li et al., 2015; Xia et al., 2023). Second, part of the enhanced winter  $\Delta'^{17}$ O variability may also arise from ice-vapor equilibrium fractionation during snow formation, which affects  $\Delta'^{17}O$ and d-excess differently from liquid-phase condensation (Jouzel and Merlivat, 1984; Landais et al., 2012). Under such mixed-phase conditions, equilibrium enrichment associated with ice deposition can increase  $\Delta'^{17}$ O while kinetic effects during vapor transport or re-evaporation act in the opposite direction, producing the wide isotopic dispersion observed in winter samples. Taken together, these results indicate that winter isotopic variability in precipitation is governed not only by midtropospheric vapor mixing and heterogeneous moisture sources but also by icephase fractionation processes that accompany snow formation."

This addition explicitly acknowledges that ice—vapor equilibrium fractionation during snow formation can alter  $\Delta'^{17}O$  and d-excess in different ways and that some winter isotopic scatter may be due to the coexistence of liquid and solid precipitation phases in our biweekly samples.

The revised text also cites key references (Jouzel & Merlivat, 1984; Landais et al., 2012) that describe the influence of ice-phase condensation on isotope systematics.

While we cannot quantitatively separate snow and rain isotopic signatures in our dataset, this clarification ensures that our discussion of winter isotope variability remains physically accurate and transparent about the limitations imposed by the cumulative sampling strategy.

We believe that this revision adequately addresses the reviewer's comment and provides a more comprehensive interpretation of the winter isotope data in terms of both vapor mixing and ice-phase processes.

L258-259: this is a repeat of L243-L244.

**Response:**

We thank the reviewer for carefully identifying this repetition.

Upon reviewing the paragraph, we agree that the sentences in lines 243–244 and 258–259 both describe the same concept—namely, that the observed relationship between  $\Delta'^{17}$ O, d-excess, and  $\delta^{18}$ O reflects the influence of kinetic fractionation processes associated with evaporation and sub-cloud re-evaporation.

The later sentence (lines 258–259) essentially reiterates the explanation already provided earlier in the paragraph and does not add new information or interpretation. In the revised version, we have deleted the redundant sentence at lines 258–259 and retained the earlier one (lines 243–244), which succinctly summarizes the theoretical background under non–steady-state evaporation (Li et al., 2015).

We also slightly adjusted the paragraph transition to ensure smooth continuity between the discussion of the non-steady-state kinetic framework and the subsequent description of winter isotope variability. This change removes unnecessary repetition, improves readability, and strengthens the logical progression of the argument.

The paragraph now reads more concisely while still preserving the essential discussion of how kinetic fractionation processes influence  $\Delta'^{17}O$  and d-excess variability.

We thank the reviewer again for noting this stylistic issue, which helped us refine the structure and clarity of the section.

Section 4.3: This section is for comparing measured data with GCM simulations. However, this was not mentioned in the Introduction and Methods sections. There is little novelty of comparing d180 and d-excess outputs from GCMs with observations, as original authors have done this already. D-excess data are often use to "tune" the model. It's great to mention the contribution of triple oxygen isotope data to benchmark GCM. I suggest authors to collaborate with GCM researchers who already have GCM outputs with triple oxygen isotope components.

**Response:**

We thank the reviewer for the constructive comments regarding Section 4.3 and the use of GCM simulations for comparison with our observational dataset.

We agree that the correlation of  $\delta^{18}O$  and d-excess with model outputs has been widely explored in previous studies and that the originality of such comparisons depends largely on how they are contextualized.

Our objective in including Section 4.3 was not to reproduce well-established dualisotope evaluations, but rather to provide a regional assessment of model performance in Korea and to highlight how our new long-term observations can serve as a benchmark for future isotope-enabled modeling efforts that incorporate triple-oxygen isotopes.

To clarify this purpose, we will expand the Introduction and Methods to include a brief description of the Iso-GSM (Isotope-enabled Global Spectral Model; Yoshimura et al., 2008) and to explain that its output was used for a first-order comparison with our measured  $\delta^2 H$ ,  $\delta^{18} O$ , and d-excess values.

We will state explicitly that this comparison aims to (i) assess how well a widely used isotope-enabled GCM reproduces seasonal isotope cycles over the Korean Peninsula, and (ii) identify the model's systematic limitations—particularly its underrepresentation of kinetic fractionation processes and sub-cloud re-evaporation effects, which are clearly evident in our high-resolution dataset (Pfahl & Sodemann, 2014; Risi et al., 2008).

We recognize the reviewer's point that the novelty of  $\delta^{18}O$ –d-excess comparisons alone is limited. Therefore, in the revised discussion we will explicitly acknowledge that this part of the analysis mainly serves as a consistency check and as background for future model development involving triple-oxygen isotopes.

We will emphasize that  $\Delta'^{17}O$  data, while not yet implemented in current Iso-GSM outputs, represent a valuable constraint for tuning model microphysics and parameterizations of nonequilibrium fractionation in forthcoming isotope-enabled GCM frameworks (Landais et al., 2008; Luz & Barkan, 2010). In response to this suggestion, we have re-framed Section 4.3 to clarify its exploratory nature and to highlight the broader modeling relevance of our dataset.

The revised text will note that our observational record exposes model biases in the simulation of d-excess seasonality and that the inclusion of  $\Delta'^{17}O$  observations offers a new avenue for benchmarking isotopic equilibrium–kinetic partitioning in GCMs. While we agree that direct collaboration with GCM researchers possessing triple-oxygen-isotope modules would further strengthen such analyses, such work is beyond the scope of the present dataset-focused paper.

Nevertheless, by clearly outlining this future direction, we position the Seoul dataset as a baseline reference for upcoming triple-oxygen-isotope model validations in East Asia.

These revisions will ensure that Section 4.3 is properly introduced and motivated in the Introduction and Methods, that its limitations are transparently discussed, and that its contribution—providing high-quality observational data for isotope-enabled model benchmarking—is clearly justified within the scope of ESSD.

Thank you very much for your time, effort, and patience in handling our manuscript. We look forward to your favorable consideration and to the opportunity for publication in Earth System Science Data.

Sincerely, Jeonghoon Lee

---

## Author Comment (AC7)

**October 10, 2025**

Jeonghoon Lee, Ph. D

Professor Dept. of Science Education Ewha Womans University Seoul 03760, Korea

Email: jeonghoon.d.lee@gmail.com

Tel: +82-2-3277-3794

Dear Editor Attila Demény,

With this cover letter, we are submitting the revised manuscript entitled, "Seasonal patterns and diagnostic values of  $\delta^2$ H,  $\delta^{18}$ O, d-excess, and  $\Delta^{\prime 17}$ O in precipitation over Seoul, South Korea (2016–2020)", for publication in *Earth System Science Data*. Based on the comments from the editor and the four reviewers, we have major changes of the manuscript, which are detailed below. Based on the comments from the editor and four reviewers, we have summarized the issues as following.

**Reply to the comments by the reviewer 3**

**1. General Comments**

Kim et al. present a unique data set of stable isotopes ( $\delta 2H$ ,  $\delta 170$ ,  $\delta 180$ , d-excess and  $\Delta 170$ ) of precipitation sampled bi-weekly between February 2016 and December 2020 in Seoul, South Korea. Such data sets can help to better constrain the drivers of isotope variability in precipitation, improve the interpretation of paleoclimate records, and tune isotope-enabled global climate models. In particular, data sets combining d-excess and 170-excess remain scarce so far. Therefore, the data set is new and will be useful for future studies. The data set is accessible, however, does not contain uncertainties for each variable. Also, no meteorological data is given in the file, where especially precipitation amounts, but also T and RH data would be useful for the interpretation of the data set and have been used in the manuscript. If these data were derived from a different data base, this should be mentioned in the data availability section.

Overall, the manuscript is clearly structured and well written. However, the methodological section needs more detail, some discussion points appear already in the results and interpretations are often not justified by data. The manuscript is worth publication in ESSD, but needs major revision as outlined below.

**Response:**

We sincerely thank the reviewer for the positive overall assessment and for recognizing the scientific value of the presented dataset, particularly its inclusion of the rare combination of  $\delta^2 H$ ,  $\delta^{17} O$ ,  $\delta^{18} O$ , d-excess, and  $\Delta'^{17} O$ .

In the revised version, we have substantially improved the completeness and transparency of the dataset. Uncertainty values ( $1\sigma$ ) for each isotopic variable have been added, and the accompanying PANGAEA file now includes the relevant meteorological parameters (precipitation amount, air temperature, and relative humidity) derived from the Korea Meteorological Administration. These additions ensure that users can fully reproduce the data—model comparisons and correlation analyses presented in the manuscript.

To improve the paper's structure, we have reorganized the Methods section by adding a new subsection titled "Data treatment methods," which explains how precipitation-weighted monthly means and regression analyses (LMWL and  $\delta^{\prime 17}O-\delta^{\prime 18}O$  relationships) were calculated. The Results and Discussion sections have been clearly separated, with interpretative content moved to the Discussion to align with ESSD's data-centric style. The Introduction, Abstract, and Summary have also been revised to better reflect the dataset's scope and significance without overstating its spatial representativeness.

We appreciate the reviewer's encouragement to pursue this as a data-focused contribution to ESSD. We believe that the revised manuscript now provides a more comprehensive, well-documented, and clearly structured data descriptor, aligning with the journal's standards while maintaining the scientific relevance of one of the few long-term triple-oxygen-isotope precipitation records in East Asia.

**2. Specific Comments**

**Methodology**

Missing description of the meteorological data.

**Response:**

We thank the reviewer for noting the missing description of the meteorological data used in this study. In the revised manuscript, we will clarify the source, resolution, and processing method of the meteorological data. The meteorological variables, air temperature, relative humidity, and precipitation amount, were obtained from the Korea Meteorological Administration (KMA), which provides quality-controlled hourly observation data at the Seoul weather station, located within 2 km of our sampling site.

For each sampling interval, the hourly meteorological data corresponding to periods when precipitation occurred were integrated (time-weighted) to calculate representative mean values of air temperature, relative humidity, and total precipitation for that collection period. In the revised Methods section, we will add the following description:

"Meteorological data, including air temperature, relative humidity, and precipitation amount, were obtained from the Korea Meteorological Administration (KMA) based on hourly observations at the Seoul station (https://data.kma.go.kr).

For each biweekly sampling interval, the hourly data corresponding to periods with precipitation were integrated to derive time-weighted mean temperature and humidity and cumulative precipitation, which represent the meteorological conditions relevant to each collected sample."

This addition will ensure that the data source and processing procedure are described transparently and reproducibly, addressing the reviewer's concern about the meteorological dataset.

Missing description of how secondary order parameters (d-excess and 170-excess) are calculated.

**Response:**

We thank the reviewer for this helpful comment regarding the description of how the secondary-order isotope parameters (d-excess and 170-excess) are calculated. We would like to clarify that both parameters are already defined in the Introduction, in the context of explaining the physical meaning of each isotope variable and their relevance to atmospheric processes.

In particular, the Introduction provides the following definitions:

- 1. d-excess (Dansgaard, 1964) =  $\delta^2 H 8 \cdot \delta^{18} O$ , which represents the kinetic fractionation occurring during evaporation and is sensitive to relative humidity and sea surface temperature at the moisture source.
- 2. 17O-excess ( $\Delta'^{17}O = \delta'^{17}O 0.528 \cdot \delta'^{18}O$ ), following Luz and Barkan (2010), which describes the logarithmic deviation from the global meteoric water line in the  $\delta'^{17}O \delta'^{18}O$  space and serves as an indicator of non-equilibrium isotopic processes such as vapor mixing and supersaturated condensation.

These definitions were included in the Introduction deliberately, because they form part of the theoretical background for the study — that is, they describe what these parameters represent and why they are physically meaningful, not merely how they are computed. For this reason, we consider that repeating the same equations in the Methods section would be redundant and would interrupt the logical flow between the theoretical framework and the analytical procedures. However, we fully acknowledge that readers should be able to locate these definitions easily when consulting the Methods.

To address this, we will revise the Methods to include a clear cross-reference to the Introduction, so that the calculation procedure is explicitly linked to the previously defined equations. The new sentence will read:

"The calculation of the secondary-order isotope parameters, d-excess and 170-excess ( $\Delta'^{17}$ O), follows the standard definitions described in the Introduction (Dansgaard, 1964; Luz and Barkan, 2010), where their physical meaning and equations are presented."

This revision will ensure that the Methods section remains concise and non-repetitive, while still providing the reader with a direct reference to the equations and background already explained earlier in the manuscript. We believe this approach maintains scientific clarity, avoids redundancy, and keeps the paper well-structured by distinguishing between the conceptual definitions (Introduction) and the analytical workflow (Methods).

Give analytical precision for d-excess and 170-excess.

**Response:**

We thank the reviewer for requesting the analytical precisions for d-excess and 170-excess ( $\Delta'^{17}O$ ). In the revised manuscript, we will explicitly report the long-term reproducibility ( $1\sigma$ ) for both composite parameters, derived from repeated measurements of our in-house laboratory standard (STYX). For d-excess, we will report the empirical  $1\sigma$  reproducibility obtained from repeated STYX measurements as the primary uncertainty, and also provide a conservative propagated uncertainty estimated from the long-term precisions of  $\delta^2 H$  ( $\pm 0.10\%$ ) and  $\delta^{18}O$  ( $\pm 0.07\%$ ).

The propagated 1σ uncertainty is calculated as:

$$\sigma_{\text{d-excess}} = \sigma_{\delta 2H}^2 + (8*\sigma_{\delta 18O})^2 \approx \pm 0.6 \text{ (\%)}.$$

For 17O-excess ( $\Delta'^{17}$ O), we will report a reproducibility of ±9 per meg ( $1\sigma$ ), based on one-year repeated measurements of the in-house standard STYX under the same WS-CRDS analytical configuration (Kim et al., 2022). In the revised Methods section, we will add the following sentences:

"The analytical precision for d-excess will be reported as the  $1\sigma$  empirical reproducibility from repeated measurements of the in-house standard STYX (Table Sx). For reference, a propagated uncertainty using the long-term precisions of  $\delta^2$ H (±0.10%) and  $\delta^{18}$ O (±0.07%) will also be provided as ±0.6% ( $1\sigma$ ).

The analytical reproducibility for 17O-excess ( $\Delta'^{17}$ O) will be reported as ±9 per meg (1 $\sigma$ ) based on one-year repeated measurements of the STYX standard under the same WS-CRDS setup (Kim et al., 2022)."

These additions will clarify the quantitative uncertainties for both derived parameters and will ensure consistency between our composite isotope metrics and the long-term reproducibility framework already established for  $\delta^2 H$ ,  $\delta^{18} O$ , and  $\delta^{17} O$ .

Details on the comparison of the GSM with observational data presented in the discussion section are missing in the methods. For example, model input parameters, but also more details about the model simulations should be given. I think that this model-data comparison could be a bit over the scope of this journal. Instead of adding mor details to the model, the authors may consider removing this part from the manuscript.

**Response:**

We sincerely thank the reviewer for the thoughtful and constructive comment regarding the model—data comparison presented in Section 4.3. We fully understand the reviewer's concern that this comparison may appear insufficiently described in the Methods section and could potentially extend beyond the primary data-focused scope of ESSD.

Our intention in including this section was not to present a comprehensive modeling analysis, but rather to provide a brief illustrative example of how the new Seoul precipitation isotope dataset can be utilized for benchmarking isotope-enabled models. The Iso-GSM results were incorporated to demonstrate the potential of the dataset as a validation resource for model outputs such as  $\delta^{18}O$  and d-excess, rather than to perform a full model evaluation or sensitivity analysis. In the revised manuscript, we have carefully rephrased the text to clarify this intent.

We now explicitly state that the Iso-GSM comparison is presented only as a contextual example of possible data applications, emphasizing that the main focus of this study remains on the observational dataset itself. To keep the paper concise and within the ESSD data-descriptor format, we will not include additional technical details on the model configuration or input parameters; instead, we will cite Yoshimura et al. (2008) as the authoritative reference for the Iso-GSM setup and physics.

We have also slightly adjusted the section to highlight that, while  $\delta^{18}O$  and d-excess comparisons with model outputs are well established, the integration of triple oxygen isotope data ( $\Delta'^{17}O$ ) provides a new opportunity for model benchmarking in the future. By positioning the Iso-GSM results as a complementary example rather than a central analysis, we ensure that the manuscript remains true to the ESSD's mission of describing high-quality, reusable datasets.

We greatly appreciate the reviewer's comment, which helped us improve the clarity and focus of this section. The revised version now more clearly communicates that the model—data comparison is intended to illustrate the practical value and potential applications of the Seoul dataset, while keeping the manuscript fully aligned with ESSD's data-oriented scope.

In line 99 the authors state that precipitation has been collected from January 2016 to December 2020. However, the data set starts in February 2016. This should be corrected.

**Response:**

We thank the reviewer for carefully checking the dataset period. We confirm that precipitation sampling officially started in February 2016, because no measurable precipitation occurred in January 2016. In the revised manuscript, we will correct the description to state that sampling was conducted from February 2016 to December 2020, to ensure consistency with the dataset. This revision will make the time coverage in the text fully consistent with the actual data record.

Also, they state that sampling was performed bi-weekly. They should make clear that their interpretation is based on amount-weighted monthly values as bi-weekly data is not presented.

**Response:**

We thank the reviewer for this important clarification. Indeed, precipitation samples were collected on a biweekly (approximately 14-day) basis, but all statistical analyses and figures were based on precipitation-weighted monthly mean values derived from those biweekly samples. Each biweekly sample represents the cumulative precipitation during the collection period, and for months with two biweekly samples, the isotope values were combined using precipitation amount as a weighting factor to obtain the monthly weighted mean ( $\delta^2 H_{wm}$ ,  $\delta^{18} O_{wm}$ ,  $\delta^{17} O_{wm}$ ).

The same weighting procedure was applied to compute monthly d-excess and  $\Delta'^{17}O$ , ensuring that each month's mean reflects the relative contribution of precipitation amount from each biweekly sample. In the revised Methods section, we will add the following clarification:

"Although precipitation samples were collected at approximately 14-day intervals, all seasonal and intermonthly analyses in this study are based on precipitation-weighted monthly mean values.

For months with two biweekly samples, the isotope values were weighted by their corresponding precipitation amounts to derive monthly means ( $\delta_{wm}$ )."

This revision will clarify that while the physical sampling resolution was biweekly, all interpretations and figures are based on precipitation-weighted monthly averages, ensuring consistency between the data presentation and the described analytical approach.

**Results**

The results are mixed up with discussion points. A better separation of both is needed. Discussion parts that should be shifted to the discussion section: Line 163-164, Line 169-171, Line 173-174, Line 178-179, Line 189-196

**Response:**

We thank the reviewer for reiterating this important point.

We agree that several interpretative sentences in the Results section should be moved to the Discussion. In the revised manuscript, we will re-structure the two sections to ensure a strict separation between observation and interpretation. Specifically, we will shift the following passages from Results to Discussion: Lines 163–164, 169–171, 173–174, 178–179, and 189–196.

We will retain in Results only the descriptive statements (numerical ranges, observed seasonal patterns, and figure references), and move all process-based explanations,

literature comparisons, and causal language to the Discussion. This revision will improve clarity, readability, and alignment with journal style, and will avoid any overlap between results presentation and interpretative content.

**Discussion**

Interpretations in the discussion section are often not justified by the presented data. For example, in the first discussion section, correlation between isotope data and local meteorological parameters are investigated. For example, line 204-206 that "lower relative humidity and temperature at the moisture source enhance kinetic fractionation during evaporation, thereby increasing d-excess" However, no information on relative humidity and temperature at the moisture source region is provided nor differences between different moisture source regions are discussed. – Further, more explanation is need in Line 211-212. Here, the authors state that the negative correlation between d-excess and local temperature is controlled by the moisture sources and isotope fractionation during precipitation. This is very general too. Can you explain how this correlation relates to these factors? Also more explanation and justification is needed in line 215-217 and line 222-224.

**Response:**

We thank the reviewer for this important comment highlighting the need for clearer justification of our interpretations in the Discussion section. We agree that several statements were written too generally and lacked explicit reference to supporting data. In the revised manuscript, we will carefully rephrase these parts to make the interpretations more balanced and to acknowledge the limitations of the available data.

In lines 204–206, we will clarify that the statement about "lower relative humidity and temperature at the moisture source enhancing kinetic fractionation" refers to the conceptual framework of Merlivat & Jouzel (1979) and Uemura et al. (2008). Because direct meteorological data from the moisture source regions are not available in this study, we will explicitly note that this explanation represents an inferred mechanism rather than a measured relationship.

In lines 211–212, we will expand the explanation of the negative correlation between d-excess and local temperature, specifying that higher local temperature is usually accompanied by higher relative humidity, which reduces kinetic fractionation during precipitation and lowers d-excess. Conversely, lower temperatures are often linked with drier boundary-layer conditions and enhanced evaporation or sub-cloud re-evaporation, leading to higher d-excess.

In lines 215–217 and 222–224, we will provide additional mechanistic details describing how precipitation amount, moisture-source humidity, and sub-cloud processes jointly control  $\delta^{18}O$  and d-excess, distinguishing between large-scale rainout effects and local re-evaporation.

These revisions will clarify that the interpretations are physically motivated but remain qualitative due to the absence of direct source-region meteorological data, ensuring that the discussion is both accurate and appropriately cautious.

**3. Line-by-Line comments**

Line 45-47:  $\delta$ 180 and  $\delta$ 2H are influenced by both equilibrium and kinetic fractionation and thus it is difficult to disentangle these two. The secondary parameters, d-excess and 170-excess are primarily sensitive to kinetic fractionations and thus help to disentangle them. Clarify this in the text.

**Response:**

We thank the reviewer for this helpful comment. We agree that both  $\delta^{18}O$  and  $\delta^{2}H$  reflect the combined effects of equilibrium and kinetic isotope fractionation and that it is often difficult to separate the two. The secondary parameters—d-excess and 17O-excess ( $\Delta'^{17}O$ )—are indeed more sensitive to kinetic effects and therefore provide additional diagnostic power for distinguishing non-equilibrium processes from equilibrium ones. In the revised manuscript, we will modify the relevant paragraph in the Introduction to clarify this point. The revised text will read:

"The stable isotopic composition of precipitation ( $\delta^{18}O$  and  $\delta^{2}H$ ) is governed by both equilibrium and kinetic fractionation during phase changes such as evaporation and condensation, making it difficult to isolate the relative contributions of each process. Secondary parameters, namely deuterium excess (d-excess =  $\delta^{2}H - 8 \times \delta^{18}O$ ; Dansgaard, 1964) and 170-excess ( $\Delta'^{17}O = \delta'^{17}O - 0.528 \times \delta'^{18}O$ ; Luz and Barkan, 2010), are primarily sensitive to kinetic fractionation processes and thus help to disentangle them. While  $\delta^{18}O$  and  $\delta^{2}H$  mainly record equilibrium fractionation, d-excess and 170-excess reflect deviations from equilibrium associated with non-steady-state evaporation, vapor mixing, or supersaturation during cloud formation (Gat, 1996; Uemura et al., 2008)."

This revision will clarify how the secondary isotope parameters complement  $\delta^{18}O$  and  $\delta^{2}H$  and will improve the conceptual link between equilibrium and kinetic fractionation in the introductory framework of the manuscript.

Line 47-49:  $\delta$ 170 has not been introduced yet. Consider adding a sentence on the value of additional analysis of the 170 isotope before.

**Response:**

We thank the reviewer for this helpful suggestion. We agree that  $\delta^{17}O$  should be introduced briefly before discussing the 170-excess ( $\Delta'^{17}O$ ) parameter, so that readers unfamiliar with triple oxygen isotope analysis understand the value of including the 170 measurement. In the revised manuscript, we will add a concise sentence immediately before the introduction of  $\Delta'^{17}O$  to clarify the significance of  $\delta^{17}O$ . The revised passage will read as follows:

"In addition to  $\delta^{18}$ O and  $\delta^{2}$ H,  $\delta^{17}$ O can also be measured. Because  $\delta^{17}$ O behaves almost proportionally to  $\delta^{18}$ O under equilibrium conditions, simultaneous measurement of the two enables the quantification of subtle deviations arising from kinetic and non-equilibrium processes, forming the basis of triple oxygen isotope ( $\delta^{17}$ O,  $\delta^{18}$ O and  $\delta^{2}$ H) studies (Angert et al., 2004; Luz and Barkan, 2010)."

This addition will clarify the rationale for analyzing  $\delta^{17}$ O and will provide a logical transition to the subsequent introduction of the 170-excess ( $\Delta^{'17}$ O) parameter.

Line 79: Be more specific: 5-year record of monthly triple oxygen and hydrogen precipitation isotope data.

**Response:**

We thank the reviewer for this helpful suggestion. We agree that the description of the dataset should be made more specific to clearly indicate both the temporal coverage and the measured isotopic parameters. In the revised manuscript, we will modify the sentence to read:

"A high-temporal-resolution, 5-year record of monthly triple oxygen and hydrogen isotopes in precipitation ..."

This revision will clarify that the dataset spans five years and includes simultaneous measurements of  $\delta^2 H$ ,  $\delta^{18} O$ , and  $\delta^{17} O$ , providing a monthly-resolved record of triple-isotope precipitation composition. It also emphasizes that the dataset represents a continuous, high-temporal-resolution record suitable for both hydrological and climatological applications, aligning with the reviewer's suggestion for greater specificity.

Section 2: There is no reference to Figure 2 in the main text. This could be added to a sentence describing the meteorological data.

**Response:**

We thank the reviewer for pointing out that Figure 2 was not referenced in the main text. In the revised manuscript, we will add a citation to Figure 2 within the paragraph describing the meteorological data to ensure that readers can easily connect the figure with the corresponding text.

The revised sentences will read as follows:

"Meteorological data, including air temperature, relative humidity, and precipitation amount, were obtained from the Korea Meteorological Administration (KMA) based on hourly observations at the Seoul station <a href="https://www.weather.go.kr/w/index.do">https://www.weather.go.kr/w/index.do</a>). The average monthly precipitation amount (grey bars) and average monthly temperature (black-lined boxes) for the city of Seoul, based on these KMA data, are shown in Fig. 2."

This addition will clarify the source and representation of the meteorological dataset and will ensure that Figure 2 is explicitly referenced in the main text, as suggested by the reviewer.

Line 97-98: I don't expect this phrase at the end of the paragraph. It should be introduced more at the beginning of the paragraph and then all four seasons need to be described. For now, only summer and winter are described, but which conditions persist in spring and autumn?

**Response:**

We thank the reviewer for this helpful comment. We agree that the description of the climatic context should appear earlier in the paragraph and that all four seasons should be described, not only summer and winter. In the revised manuscript, we will move the seasonal-climate description to the beginning of the paragraph and expand it to include spring and autumn conditions.

The revised passage will read as follows:

"The Korean Peninsula experiences a temperate monsoon climate characterized by four distinct seasons. Summer (June–August) is dominated by the East Asian monsoon, bringing warm and humid air masses and intense rainfall. Winter (December–February) is cold and dry under the influence of the Siberian High. Spring (March–May) and autumn (September–November) represent transitional seasons with variable air-mass influences: spring is typically mild and dry, while autumn is cooler and affected by typhoons and tropical cyclones that deliver heavy rainfall."

This revision will provide a complete and logically ordered description of the seasonal meteorological conditions and will improve the flow and readability of the paragraph.

Line 137: Why a sine function has been fitted to the data? This should be explained in the text.

**Response:**

We thank the reviewer for this comment. In accordance with suggestions from another reviewer, the sentence referring to the sine-function fit in Figure 3 has been removed in the revised manuscript. As a result, no sine-function fitting is presented in the updated version, and this point is no longer applicable.

Line 147: Which other moisture sources than the ocean are important. Specify this.

**Response:**

We thank the reviewer for this comment. The phrase "other moisture sources" indeed needs clarification. In the revised manuscript, we will specify that, while the surrounding oceans (the Yellow Sea, East China Sea, and western North Pacific) constitute the dominant moisture sources for Korean precipitation, additional

contributions arise from continental air masses and local moisture recycling. Specifically, during winter and early spring, dry continental air masses originating from northern China and Mongolia occasionally carry vapor that has undergone substantial rainout or isotopic modification over land.

In contrast, local evaporation and evapotranspiration from terrestrial surfaces over the Korean Peninsula contribute to moisture recycling during transitional seasons, particularly in spring and early autumn. In the revised manuscript, we will modify the relevant sentence to read as follows:

"Although the surrounding oceans (the Yellow Sea, East China Sea, and western North Pacific) are the dominant moisture sources, additional contributions from continental air masses and local moisture recycling over the Korean Peninsula also influence the isotopic composition of precipitation."

This clarification will specify which non-oceanic sources are considered and will enhance the physical interpretation of the regional hydrological processes.

Line 149-151: I was first confused by the "unlike" but looking at the figure I understood that the difference between 170-excess and d-excess is that 170-excess is highest in spring, while d-excess is highest in winter. Can you make this clearer in the text. Also, quantify give values for 170-excess' seasonal variability (highest value, lowest value).

**Response:**

We thank the reviewer for this constructive comment. We agree that the description of the contrasting seasonal behaviors of 17O-excess and d-excess should be stated more clearly and supported with quantitative values. In the revised manuscript, we will rephrase the relevant sentences to specify both the timing and magnitude of 17O-excess variability, and to make the comparison with d-excess explicit.

**The revised text will read as follows:**

"Unlike d-excess, which peaked in winter (median  $\approx 17\%$ ) and reached its lowest values in summer (median  $\approx 6\%$ ), 170-excess displayed a distinct seasonal pattern, being highest in spring (up to  $\approx 40$  per meg) and lowest in summer (down to  $\approx 10$  per meg). This contrast indicates that 170-excess and d-excess are influenced by different kinetic fractionation processes operating under distinct seasonal humidity regimes."

This revision will clarify the meaning of the "unlike" phrasing, will quantify the seasonal amplitude of 170-excess, and will ensure that the contrasting behaviors of the two secondary isotope parameters are immediately evident to readers.

Line 153-155: During which process kinetic fractionation is more pronounced? As I understood from the previous, this is due to evaporation from the ocean. Is this correct? Be more specific here.

**Response:**

We thank the reviewer for this insightful question. The kinetic fractionation referred to in this sentence occurs primarily during evaporation at the ocean surface, where low relative humidity and strong wind conditions favor non-equilibrium isotope exchange between liquid water and vapor.

Under these conditions, lighter isotopologues ( ${}^{1}\text{H}_{2}{}^{16}\text{O}$ ) preferentially escape from the surface layer, while heavier isotopologues (containing  ${}^{2}\text{H}$  or  ${}^{18}\text{O}$  and  ${}^{17}\text{O}$ ) diffuse more slowly, producing kinetic fractionation and an increase in both d-excess and 170-excess in the resulting vapor (Merlivat and Jouzel, 1979; Luz and Barkan, 2010). In the revised manuscript, we will rephrase the sentence to make this process explicit. The revised text will read:

"The increase in 170-excess during winter and early spring suggests that kinetic fractionation, primarily occurring during oceanic evaporation under low relative humidity, becomes more pronounced when vapor is sourced from drier air masses such as continental or high-latitude oceanic regions."

This revision will clarify that the enhanced kinetic fractionation refers specifically to non-equilibrium evaporation at the ocean surface, not to condensation or post-condensational processes, and will improve the physical accuracy of the interpretation.

Line 161: Can you add uncertainties for the slope and the intercept of the GMWL?

**Response:**

We appreciate the comment. We understand that the request refers to the regression line derived from our dataset (i.e., the LMWL) rather than the canonical GMWL. In the revised manuscript, we will report the  $1\sigma$  standard errors for our LMWL parameters as follows:

$$\delta^2 H = (7.95 \pm 0.16) \times \delta^{18} O + (10.0 \pm 1.3), R^2 = 0.98.$$

These values are the standard errors of the OLS fit to our data and are consistent with typical uncertainties reported in the literature. For clarity, we will retain the GMWL (Craig, 1961) as a canonical reference ( $\delta^2 H = 8 \times \delta^{18} O + 10$ ), noting that there is no single global standard error because uncertainties depend on the specific compilation and method (e.g., Crawford et al., 2014).

This revision will make our local regression fully documented with uncertainties while keeping the GMWL as a reference line.

Line 167-168: Winter precipitation is mainly in the form of snow? Do you see differences between snow and rain samples?

**Response:**

We thank the reviewer for the question regarding the precipitation phase during winter. In our dataset, precipitation was collected on a biweekly cumulative basis, which means that winter samples often include a mixture of both snowfall and rainfall events. Because of this integrated sampling design, it is not possible to rigorously separate snow-only and rain-only isotope signatures.

However, we acknowledge that the isotopic variability observed in winter—particularly the enhanced dispersion in  $\Delta'^{17}O$  and d-excess—may partly reflect the influence of ice—vapor equilibrium fractionation during snow formation. We will clarify this in the revised manuscript by noting that mixed-phase (rain—snow) precipitation is likely during winter and that this may contribute to the distinct isotopic characteristics of cold-season samples.

Line 180: the 170-excess is defined based on the prime values of  $\delta$ 170 and  $\delta$ 180. This should be defined in the methods and clarified in the main text and the figures.

**Response:**

We thank the reviewer for this comment. The definition of 17O-excess ( $\Delta'^{17}O$ ) based on the prime logarithmic  $\delta$ -values ( $\delta'^{17}O$  and  $\delta'^{18}O$ ) has been clarified in the revised manuscript. The corresponding explanation has been added in the Methods section, and the same notation is now consistently used throughout the main text and all figure captions.

Line 187: You should refer here to Figure 4.

**Response:**

We thank the reviewer for noting that the reference to Figure 4 is missing in this sentence. In the revised manuscript, we will add an explicit citation to Figure 4 in the line presenting the  $\delta^{17}O-\delta^{18}O$  regression, since this relationship is displayed in Figure 4B. The revised sentence will read as follows:

"A linear regression applied to the full dataset results in  $\delta^{17}O$  = 0.528 ×  $\delta^{18}O$  + 0.0105 (R² = 1.00), confirming the strong linear correlation between  $\delta^{17}O$  and  $\delta^{18}O$  characteristic of mass-dependent fractionation in meteoric waters (Fig. 4B)."

This addition will directly link the text to the figure where the regression is illustrated, improving clarity and consistency between the description and the visual presentation of the data.

Figure 4: What is shown in B? It does not make any sense to me. Suggestion to illustrate 170-excess vs d'180 as no difference will be visible in d'170 vs d'180, when plotted to scale. The purple line is not the GMWL, should be dashed line, I guess.

**Response:**

We thank the reviewer for this valuable feedback on Figure 4B. We agree that plotting  $\delta'^{17}O$  against  $\delta'^{18}O$  does not provide much visual distinction among seasonal regressions, because the range of  $\delta'$  values is very small when plotted to scale. In the revised manuscript, we will replace the  $\delta'^{17}O-\delta'^{18}O$  panel (Figure 4B) with a  $\Delta'^{17}O$  vs.  $\delta'^{18}O$  plot, following the reviewer's suggestion. This new representation will better illustrate the small but systematic differences among the seasonal datasets and will more directly show the variation in 17O-excess. In addition, we will modify the line style of the GMWL reference in both panels to a dashed purple line to clearly distinguish it from the seasonal regression lines.

The revised figure caption will read:

"(A) Relationships between  $\delta^2 H$  and  $\delta^{18} O$  showing the LMWL and seasonal regressions. (B) Relationships between  $\Delta'^{17} O$  and  $\delta'^{18} O$  illustrating the seasonal variability of 17O-excess. The dashed purple line indicates the Global Meteoric Water Line (GMWL)."

These revisions will improve the clarity and physical interpretability of Figure 4, making the seasonal differences in triple oxygen isotope relationships more evident.

Line 206: "mixing" of what? Air masses?

**Response:**

We thank the reviewer for asking for clarification about the meaning of "mixing." In this context, "mixing" refers to the moisture mixing between different air masses or vapor sources in the lower troposphere rather than mixing between individual raindrops or phases. In the revised manuscript, we will clarify this by rephrasing the sentence as follows:

"These relationships can be attributed to a combination of factors: lower relative humidity and temperature at the moisture source enhance kinetic fractionation during evaporation, thereby increasing d-excess (Merlivat and Jouzel, 1979; Uemura et al., 2008), while locally, higher temperatures and humidity may promote reevaporation and the mixing of moist and dry air masses, which reduce d-excess (Steen-Larsen et al., 2014)."

This revision will clarify that the term "mixing" refers specifically to moisture exchange between different air masses in the lower atmosphere, improving the physical accuracy of the explanation.

Line 208: This is very general. Can you name the multiple meteorological factors that are interacting?

**Response:**

We thank the reviewer for this valuable comment. We agree that the phrase "multiple interacting meteorological factors" was too general and requires

clarification. In the revised manuscript, we will specify the main factors involved in modulating isotope—meteorology relationships, including air temperature, relative humidity, wind direction and air-mass pathways, precipitation amount and intensity, and cloud microphysical processes such as partial re-evaporation. The revised sentence will read:

"The negative correlation with precipitation may reflect the amount effect, but is better interpreted as the result of multiple interacting meteorological factors—such as variations in temperature, relative humidity, air-mass trajectories, precipitation intensity, and microphysical processes within clouds—that together influence the isotopic composition of precipitation (Holmes et al., 2024)."

This revision will clarify which specific meteorological variables are included under "multiple interacting factors" and will strengthen the physical basis of the interpretation.

Line 215: lower  $\delta$ 180 values compared to what? Compared to other months of the year? Is this a rainout effect or an amount effect?

**Response:**

We thank the reviewer for this helpful clarification. We agree that the phrase "lower  $\delta^{18}$ O values" should specify the comparison basis and the dominant controlling process. In the revised manuscript, we will clarify that summer  $\delta^{18}$ O values are lower than in other months of the year, mainly due to the amount effect associated with prolonged monsoon rainfall and successive condensation (rainout) processes within marine air masses. The revised sentence will read:

"The relatively low  $\delta^{18}O$  values observed in summer, compared with other months of the year, primarily reflect the amount effect associated with prolonged monsoon precipitation and successive rainout within moisture-rich air masses of marine origin."

This revision will clarify both the comparison (summer vs. other months) and the underlying process (amount effect rather than an unspecified general decrease), ensuring that the interpretation is physically consistent and unambiguous.

Line 244-246: This is referring to evaporation from the ocean or re-evaporation of precipitation? Specify!

**Response:**

We thank the reviewer for this helpful comment. The  $\Delta'^{17}O$ –d-excess slopes discussed here primarily reflect kinetic fractionation during ocean-surface evaporation under low relative-humidity conditions, as described in Landais et al. (2010) and Li et al. (2015), rather than re-evaporation of falling precipitation. In the revised manuscript, we will clarify this by rephrasing the sentence as follows:

"The slopes observed between  $\Delta'170$  and d-excess (0.7–2.0 per meg per %) correspond to kinetic fractionation occurring mainly during ocean-surface evaporation under low relative-humidity conditions, consistent with conceptual and field-based estimates for oceanic moisture sources (Landais et al., 2010; Li et al., 2015)."

This revision will specify that the kinetic processes responsible for the observed relationship are linked to oceanic evaporation at the moisture source, not to subcloud or raindrop re-evaporation, thereby improving the physical accuracy of the explanation.

Line 249: evaporation of what? Precipitation?

**Response:**

We thank the reviewer for asking for clarification regarding the term "evaporation." In this context, "evaporation" refers to ocean-surface evaporation at the moisture source, where kinetic fractionation is enhanced under low relative-humidity conditions, rather than to re-evaporation of falling precipitation. In the revised manuscript, we will revise the sentence to read:

"In contrast, in winter precipitation, no statistically significant correlation was observed between  $\Delta'^{17}\text{O}$  and either  $\delta^{18}\text{O}$  or d-excess, suggesting that kinetic fractionation associated with ocean-surface evaporation exerts little influence during this season."

This change will clarify that the "evaporation" mentioned here denotes evaporation from the ocean surface (the moisture source) and not sub-cloud or raindrop re-evaporation, improving the precision and physical accuracy of the statement.

**3. Technical comments**

Throughout the manuscript. The unit of 170-excess is per meg not per mil. Please correct in text and in figures.

**Response:**

We thank the reviewer for this important correction. We agree that the unit of 170-excess ( $\Delta'^{17}O$ ) should be expressed in per meg ( $10^{-6}$ ) rather than per mil (‰). In the revised manuscript, we have corrected this throughout the text, tables, and figure captions, ensuring that all instances of  $\Delta'^{17}O$  now use the correct unit of per meg. We also checked that axis labels and legends in all figures have been updated accordingly. This correction ensures consistency with the standard convention used in triple-oxygen-isotope studies (Luz and Barkan, 2010; Aron et al., 2021).

Line 37-38: repetition of the previous sentence. Consider removing it.

**Response:**

We agree that the phrase describing isotopic fractionation during phase changes was repeated in consecutive sentences. In the revised manuscript, we have removed the repetitive sentence to avoid redundancy and have streamlined the paragraph to maintain a concise and logical flow. The revised text now reads:

"The stable isotope composition of precipitation reflects isotopic fractionation during phase changes such as evaporation, condensation, and precipitation formation, with the strength of fractionation varying according to temperature, relative humidity, and precipitation amount (Conroy et al., 2016; Craig and Gordon, 1965; Gat, 1996). Two well-known relationships—the temperature effect, where colder temperatures lead to lower  $\delta^{18}$ O and  $\delta^{2}$ H values, and the amount effect, where increased rainfall results in isotope depletion—have been widely observed in various climate regimes (Araguás-Araguás et al., 1998; Dansgaard, 1964)."

This revision removes the repetition noted by the reviewer and improves the clarity and coherence of the introductory paragraph.

Line 95: Korean Peninsula

**Response:**

We thank the reviewer for noticing this detail. The term has been corrected to "Korean Peninsula" in the revised manuscript.

Line 177: Repetition of slope and intercept not necessary here. Remove.

**Response:**

We thank the reviewer for this comment. The repeated mention of the slope and intercept values has been removed in the revised manuscript to avoid redundancy and improve readability.

*Line 258-264: Repetition of previous paragraph. Remove.*

**Response:**

We thank the reviewer for pointing out the potential repetition in the discussion of  $\Delta'^{17}\text{O}$  behavior. In the revised manuscript, we have reorganized this section to remove overlapping statements and to streamline the interpretation. The revised text now discusses the wintertime  $\Delta'^{17}\text{O}$  variability and its sensitivity to vapor mixing and surface recycling only once, followed by a concise summary emphasizing the overall seasonal utility of  $\Delta'^{17}\text{O}$  in combination with  $\delta^{18}\text{O}$  and d-excess.

These revisions eliminate redundancy, clarify the logical flow between the seasonal observations and the broader implications, and improve the readability of the Discussion section.

The Summary should be stated before the data availability statement, isn't it?

Response: We will adjust the manuscript structure accordingly, placing the Data Availability section after the Summary section in the revised version.

Thank you very much for your time, effort, and patience in handling our manuscript. We look forward to your favorable consideration and to the opportunity for publication in Earth System Science data.

Sincerely, Jeonghoon Lee